

**Geochronological and Geochemical Effects of Zircon Chemical Abrasion: Insights from Single-**
**Crystal Stepwise Dissolution Experiments**
**Alyssa J. McKanna[1,2], Blair Schoene[1], and Dawid Szymanowski[1,3]**
[1]Princeton University, Department of Geosciences, Princeton, New Jersey 08544, USA
[2]Los Alamos National Lab, Los Alamos, New Mexico 87545, USA
[3]Institute of Geochemistry and Petrology, ETH Zürich, 8092 Zürich, Switzerland
**Correspondence:** Alyssa J. McKanna (ajmckanna@lanl.gov)
**Abstract**
Chemical abrasion in hydrofluoric acid (HF) is routinely applied to zircon grains prior to U-Pb
dating by isotope dilution thermal ionization mass spectrometry (ID-TIMS) to remove radiation-
damaged portions of grains affected by Pb loss. Still, many chemically abraded datasets exhibit
evidence of residual Pb loss. Here we test how the temperature and duration of chemical
abrasion affects zircon U-Pb and trace element systematics in a series of 4-hour, single-crystal
stepwise dissolution experiments at 180 °C and 210 °C. Microtextural data for the zircon
samples studied is presented in a complementary paper by McKanna et al. (2023). We find that
stepwise dissolution at 210 °C is more effective at eliminating U, common Pb ($Pb_c$), and light
rare earth element (LREE) enriched material affected by open system behavior; reduces the
presence of leaching-induced artefacts that manifest as reverse discordance; and produces
more consistent and concordant results in zircon from the three rocks studied. We estimate
that stepwise dissolution in three 4 h steps is roughly equivalent to a single ~8 h leaching step
due to the insulating properties of the PTFE sleeve in the Parr pressure dissolution vessel,
whereas traditionally labs utilize a single 12-hour leaching step. To better understand the
causes of Pb-loss in zircon, we calculate time-integrated alpha dose estimates for leachates and
residues from measured radionuclide concentrations to determine: 1) the alpha dose of the
material dissolved at the two leaching conditions, and 2) the apparent minimum alpha dose
required for Pb loss to occur: $\geq 6 \times 10^{17}$ α/g. We conclude that a single 8 h leaching step at 210
˚C should yield crystallization ages in the majority of zircon and that this can be used as an
effective approach for routine analysis. However, Ultimately, the effectiveness of any chemical
abrasion protocol will be sample-dependent. By framing Pb loss and zircon solubility in terms of
alpha dose, however, workers can better tailor the chemical abrasion process to specific zircon
samples to improve the accuracy and precision of U-Pb results.
**1. Introduction**
Zircon U-Pb geochronology by isotope dilution thermal ionization mass spectrometry (ID-TIMS)
has played a pivotal role in constraining the timing and tempo of processes on Earth from the
Hadean to the Pleistocene. Zircon is a remarkable chronometer, in part because crystalline
zircon is exceptionally chemically and physically durable. The zircon structure, however, can
accumulate radiation damage over time. Radiation damage is principally caused by alpha recoil



events in the ²³⁸U, ²³⁵U, and ²³²Th decay series and the spontaneous fission of ²³⁸U (Ewing et al.,
2003; Meldrum et al., 1998; Weber, 1990). Radiation-damaged zircon can lose Pb and less
commonly U, violating the basic requirement of geochronology that neither parent nor
daughter isotopes are lost through time except through radioactive decay (Geisler et al., 2002).
Fortunately, the dual ²³⁸U/²⁰⁶Pb and ²⁰⁷Pb/²³⁵U decay schemes provide a self-check mechanism
by which open system behavior can be identified in zircons older than several hundred Ma
(Mezger and Krogstad, 1997; Corfu, 2013). In the Phanerozoic, however, the dual decay system
becomes less effective at recognizing Pb-loss, since the trajectory of Pb-loss follows Concordia,
and the precision of ²⁰⁷Pb/²³⁵U dates is also lower than corresponding ²³⁸U/²⁰⁶Pb dates due to
the shorter radioactive half-life of ²³⁵U and lower isotopic abundance (Corfu, 2013; Schoene,
52  2014).

In a seminal study, Mattinson (2005, 2011) – building off the previous findings of Krogh and
Davis (1975) and Todt and Büsch (1981) – demonstrated that the most radiation-damaged
portions of zircon can be effectively removed by hydrofluoric acid through a series of stepwise
dissolution experiments on multi-grain aliquots. He showed that early leaching steps sampled
high-U material with discordant U-Pb dates, while later leaching steps sampled low-U residues
unaffected by open-system behavior. Mattinson (2005, 2011) further established that partially
annealing zircon samples prior to leaching helps to minimize the unwanted isotopic
fractionation effects that plagued earlier leaching attempts (Davis & Krogh, 2000; Todt &
Büsch, 1981). These experimental findings revolutionized the field of zircon U-Pb
geochronology by allowing scientists to attain meaningful geochronological results from
previously unusable zircon affected by open-system behavior. Air abrasion – the pre-treatment
technique previously used to remove crystal rims (thought to be high in U) and improve U-Pb
concordance (Krogh, 1981) was largely abandoned. Today, a variation of Mattison's approach –
termed chemical abrasion – is applied to virtually all zircon grains prior to ID-TIMS U-Pb isotopic
analysis. In this variation, zircon crystals are annealed at 800 °C to 1200 °C for 36 h to 60 h and
then leached in concentrated HF at 180 °C to 210 °C for 10 h to 18 h prior to dissolution and
isotopic analysis (Mundil et al., 2004; Huyskens et al., 2016; Widmann et al., 2019).
The decrease in sample size from multi-grain aliquots to portions of single crystals and the
concurrent increase in analytical precision in TIMS over the past half-century (e.g., Schoene,
2014) demands a critical re-evaluation of the chemical abrasion technique and the accuracy of
the U-Pb ages that the Earth science community has come to rely on. Many studies have now
shown that chemically abraded zircon samples often exhibit residual Pb-loss. This challenge is
widely recognized in the ID-TIMS U-Pb community and has prompted new investigations into
the effects of different annealing and leaching conditions on geochronological outcomes
(Huyskens et al., 2016; Widmann et al., 2019), new statistical approaches for evaluating over-
dispersed U-Pb datasets (Keller, 2023), and microstructural studies of chemically abraded zircon
(McKanna et al., 2023).
We build on earlier work of Mattinson (2005, 2011) and present a series of new stepwise
dissolution experiments performed at the single-crystal scale. We evaluate the effects of
stepwise chemical abrasion at 180 °C and 210 °C on zircon U-Pb and trace element systematics



in three zircon samples – AS3, SAM-47, and KR18-04 – which span a range of crystallization
ages, geological settings, and radiation damage densities. These zircons come from the same
sample aliquots as studied by McKanna et al. (2023) in their recent microstructural
investigation of zircon dissolution which presents a unique opportunity to integrate zircon
microtextures, geochronology, and geochemistry.
**2. Methods**
Zircon samples were annealed in quartz crucibles at 900 °C for 48 h in air in a box furnace prior
to the start of the experiments. Annealed grains were mounted in epoxy, polished, and imaged
by cathodoluminescence (CL) or backscattered electron (BSE) imaging using a XL30 FEG
scanning electron microscope equipped with a mini-Gatan CL detector and a semiconductor
BSE detector housed at the PRISM Imaging and Analysis Center at Princeton University.
Representative images of AS3, SAM-47, and KR18-04 crystals can be found in McKanna et al.
(2023) Fig. 3 and Fig. 4.
The stepwise partial dissolution protocol outlined here is very similar to that of Keller et al.,
(2019, their Fig. 1). Crystals were plucked from their epoxy mounts, rinsed in 30% $HNO_3$, and
individually transferred to 200 µL PFA microcapsules for partial dissolution in ~100 µL of
concentrated HF. Microcapsules were loaded into a PTFE-lined Parr pressure dissolution vessel
with 5 mL moat HF and placed in a box oven set to 180 °C or 210 °C for a period of 4 h. At the 4
h mark, the pressure vessel was removed from the oven and placed in front of a fan to cool to
room temperature.
The microcapsules were then removed from the pressure vessel and the leachate (the dissolved
zircon-HF mixture) from each microcapsule was transferred to a clean 7 mL PFA beaker using a
pipette. A fresh, acid-cleaned pipette tip was used for each sample transfer. Approximately 100
µL of 6N HCl was added to the residue (the remaining undissolved zircon) in the microcapsule,
and the microcapsule was capped and placed on the hotplate for 1 h. The 6N HCl was then
pipetted off the residue and added to the 7 mL PFA beaker with the sample leachate. The
residue was then sequentially rinsed in the microcapsule using a pipette with 3N HCl, 6N HCl,
30 % $HNO_3$, and concentrated HF. These rinses were discarded. About 100 µL of fresh
concentrated HF was then added to each residue for the second round of step leaching. In total,
samples were partially dissolved in a series of three 4-h leaching steps generating a L1, L2, and
L3 leachate for each zircon crystal.
After the L3 leachate was collected, the residue was again rinsed with acid and ~100 µL of fresh
HF was added to the microcap. Each residue was spiked with the EARTHTIME $^{205}Pb$-$^{233}U$-$^{235}U$
tracer (Condon et al., 2015; McLean et al., 2015) and dissolved in a Parr pressure dissolution
vessel in a box oven at 210 °C for 48 to 60 h. Each leachate was spiked with the same tracer,
capped, and placed on the hot plate for the same duration. Both leachates and residues were
then dried down on the hot plate. Residues were redissolved in ~100 µL of 6N HCl in the Parr
pressure vessel in the box oven at 180 °C overnight, and leachates were redissolved in ~100 µL
of 6N HCl on the hot plate overnight. Afterward, all residues and leachates were dried down on



the hot plate and redissolved in 3N HCl in preparation for ion exchange chromatography. This
procedure was modified slightly for half the KR18-04 zircon samples step-leached at 210 °C to
evaluate whether a protocol change could mitigate unwanted U-Pb elemental fractionation
effects as suggested by Mattinson (2005) with the goal of eliminating U-loss caused by the
incomplete dissolution of fluoride salts. For these samples, after each HF leachate was
collected, zircon residues were dried down completely on the hot plate before the addition of
~100 μL of 6N HCl. Microcaps were then transferred back to the Parr pressure vessel and
redissolved at 180 °C overnight in the box oven. The 6N HCl liquid was then pipetted off the
residue and again added to the sample's HF leachate in the PFA 7 mL beaker. This procedure
was repeated for the L2 and L3 leachates. All other steps remained the same.
PTFE columns were prepared with 50 μL of Eichrom AG1-X8 anion exchange resin, cleaned, and
equilibrated. U-Pb ion exchange chemistry followed the protocol established by Krogh (1973)
and modified by Schoene et al., (2010) for the collection of trace elements. Combined U and Pb
fractions were dried down with trace 0.05 M $H_3PO_4$ and loaded onto a zone-refined Re filament
with a Si-gel emitter (Gerstenberger & Haase, 1997) for isotopic analysis on one of the two
IsotopX Phoenix TIMS at Princeton University. Pb isotopes were measured on either the
Daly/photomultiplier detector or ATONA Faraday system (Szymanowski and Schoene, 2020),
and U isotopes were measured as oxides on Faraday cups with $10^{12}$ Ω resistors or on the
ATONA Faraday system. Mass fractionation in Pb isotopic analyses was corrected for with
factors specific to each detector system, derived from a compilation of in-run values measured
in double-spiked samples; for U the correction used the known $^{233}U/^{235}U$ of the tracer. Tripoli
and ET-Redux software (Bowring et al., 2011; McLean et al., 2011) were used for processing
isotopic data and error propagation, assuming a sample $^{238}U/^{235}U$ ratio of 137.818 ± 0.045 (2σ)
(Heiss et al., 2012). All reported $^{206}Pb/^{238}U$ and $^{207}Pb/^{206}Pb$ dates are calculated using the decay
constants of Jaffey et al., (1971) and Th-corrected assuming a magma Th/U ratio of 3.5.
Reported uncertainties reflect 2σ analytical uncertainties. Common Pb corrections assume a
composition equivalent to the blank.
Major and trace element analyses were made using a Thermo Scientific iCap-Q inductively
coupled plasma mass spectrometer (ICPMS) at Princeton University following the procedure
developed by Schoene et al., (2010), with analytical parameters described in O'Connor et al.,
(2022) U concentrations were calculated from Th concentrations measured by ICPMS and the
Th/U ratio estimated from radiogenic $^{208}Pb$ and the $^{206}Pb/^{238}U$ age assuming concordance
between the U-Pb and Th-Pb systems. The percent zircon dissolved is calculated using Zr
abundances: $(Zr_{step}/Zr_{total}) \times 100$. LREE-indices (LREE-I) quantify LREE-enrichment in zircon
which can reflect chemical alteration or sample contamination. The lower the LREE-I, the higher
the LREE-enrichment. LREE-I is calculated as [Dy]/[Nd] + [Dy]/[Sm] following Bell et al., (2016).
**3. Background and results**
**3.1 AS3**
**3.1.1 Geologic setting and sample description**



AS3 zircons are from an anorthosite from the Duluth Complex of northern Minnesota, USA
which formed during the Mesoproterozoic North American Midcontinent Rift (92°09'32.4",
46°45'43.4") (Paces & Miller, 1993; Miller et al., 2002; Schmitz et al., 2003; Swanson-Hysell et
al., 2019, 2020). The Duluth Complex is a massive layered mafic intrusion. The anorthositic and
layered series of the complex were emplaced at ~1096 Ma over a duration <1 m.y. (Swanson-
Hysell et al., 2020). The voluminous magmatism that formed the Duluth Complex is attributed
to decompression melting due to lithospheric extension occurring atop an upwelling magmatic
plume (Swanson-Hysell et al., 2020). Rifting in the region ceased at ~1084 Ma (Swanson-Hysell
et al., 2019). Thermochronology data from Minnesota River Valley in southern Minnesota
suggest that rocks in the region have sat at near-surface temperature conditions since the
Neoproterozoic (Guenthner et al., 2013; McDannell et al., 2022).
The AS3 sample studied is the same as that of Takehara et al., (2018). The rock sample is
composed of plagioclase, amphibole, clinopyroxene, and ilmenite with minor K-feldspar,
apatite, zircon, and baddeleyite. Partially chloritized amphiboles, altered plagioclase, and
zeolite veins indicate that this sample of AS3 has interacted with low-temperature
hydrothermal fluids as previously described (Takehara et al., 2018). Zircon grains are large and
occur as orange-to-orangish brown tabular prisms or anhedral shards. Grains are fractured and
often have large melt inclusions oriented elongate to the *c*-axis. Crystals exhibit concentric and
convolute zonation patterns, and many grains are hydrothermally altered (McKanna et al.,
2023; Takehara et al., 2018). Included and altered grains were included in the experiments to
evaluate how well geochemical data traces the dissolution of inclusions and altered material.
Raman data indicate that grains have accumulated high radiation damage densities with
equivalent alpha doses of $2\times10^{17}$ to >$1\times10^{19}$ α/g with significant intracrystalline variations in
radiation damage (McKanna et al., 2023).

**3.1.2 Previous geochronology**

Paces and Miller (1993) presented the first U-Pb geochronological data for AS3 zircon. These
authors found that six multi-grain aliquots of air-abraded zircon crystals produced concordant
ID-TIMS U-Pb dates and assigned the sample a weighted-mean $^{207}Pb/^{206}Pb$ age of 1099.1 ± 0.5
Ma (2σ). Schmitz et al., (2003) later conducted additional ID-TIMS U-Pb isotopic analysis on
individual air-abraded AS3 zircon. The authors found that several crystals produced discordant
dates affected by recent Pb loss. Twelve concordant analyses yielded a concordia age of 1099.1
± 0.2 Ma (2σ). Eight grains from the same sample were later analyzed by chemical abrasion ID-
TIMS by Schoene et al., (2006) producing concordant dates with weighted mean $^{206}Pb/^{238}U$ and
$^{207}Pb/^{206}Pb$ ages of 1095.9 ± 0.2 Ma and 1098.6 ± 0.3 (2σ), respectively. These grains were
annealed at 900 °C for 60 h and chemically abraded in an HF-HNO₃ mixture at 180 °C for 12 to
14 h. Age differences between these and previous results were attributed by the authors to
differences in tracer calibration, which had been redone as part of (Schoene et al., 2006).
Takehara et al., (2018) later demonstrated that zircons from a different sample of AS3 collected
from the same sample locality are strongly affected by hydrothermal alteration; sensitive high-
resolution ion microprobe (SHRIMP) analyses showed that hydrothermally altered zones

Figure 1. U-Pb concordia diagrams for the 180 °C (left) and 210 °C (right) AS3 experiments. **(a)** All data are depicted except for L1 leachates with Pb*/Pb$_c$ values < 1. **(b)** Close up of L2, L3, and R data. **(c)** Close up of zircon residue data. Ellipses with dashed borders were excluded from the weighted-mean $^{206}$Pb/$^{238}$U age for the 180 °C experiment. All ellipses reflect 2σ analytical uncertainties.

yielded normally discordant U-Pb analyses, were enriched in incompatible trace elements
including LREEs, Ca, Mn, Fe, Al, Li, and K, and depleted in Zr and Si.



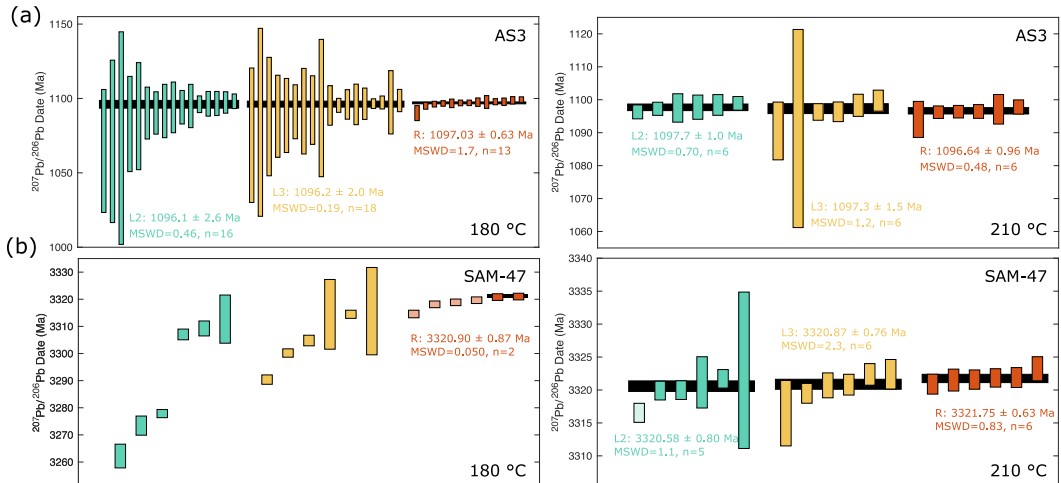

Figure 2. Ranked order $^{207}$Pb/$^{206}$Pb dates for the **(a)** AS3 and **(b)** SAM-47 experiments. Black bars represent weighted means. Bar heights and quoted uncertainties reflect propagated 2σ analytical uncertainties.

### 3.1.3 U-Pb and trace element results


ID-TIMS U-Pb results for AS3 samples are presented in Fig. 1, Fig. 2, and Table S1. L1 leachates
are younger than zircon residues indicating the dissolution of material strongly affected by Pb
loss. L1 leachates have large age uncertainties due to high amounts of Pb$_c$. Consequently, many
large error ellipses overlap the Concordia curve. L1 leachates with higher radiogenic to common
Pb (Pb*/Pb$_c$) ratios and better age precision are normally discordant. L2 and L3 leachates are
older than L1 leachates and form a discordia line through analyses that are either normally
discordant, concordant, or reversely discordant. The upper intercept ages for the discordia lines
agree within uncertainty with the weighted mean $^{206}$Pb/$^{238}$U age for residues treated at 210 °C.
The lower intercept ages of the discordia lines are zero-age. Ages for L2 and L3 leachates
treated at 180 °C are more widely dispersed than ages for L2 and L3 leachates treated at 210 °C.
The 180 °C leachates also have larger uncertainties due to higher Pb$_c$ contents.

Residues treated at 210 °C form a single, concordant age population with weighted mean
$^{206}$Pb/$^{238}$U age of 1096.42 ± 0.49 Ma (MSWD = 1.7; Fig. 1) in agreement with previous
geochronology (Schoene et al., 2006). U-Pb ages of residues treated at 180 °C are dispersed
along Concordia; a cluster of residue analyses agree with the 210 °C result yielding a weighted
mean $^{206}$Pb/$^{238}$U age of 1096.29 ± 0.36 Ma (MSWD = 2.3), but a few analyses are either older or
younger. Two of the 180 °C residues are reversely discordant. Weighted-mean $^{207}$Pb/$^{206}$Pb ages
for all leachates and residues agree within uncertainty (Fig. 2). The weighted-mean $^{207}$Pb/$^{206}$Pb
ages obtained for the 180°C and 210 °C residues are 1097.03 ± 0.63 Ma (MSWD = 1.7) and
1096.64 ± 0.96 Ma (MSWD = 0.48), respectively. These dates are slightly younger than previous
geochronology (Schoene et al., 2006).



Figure 3. Chondrite-normalized REE spider diagrams for the 180 °C (left) and 210 °C (right) AS3 experiments comparing results for leachates and residues.

Leachates from the 180 °C experiment are enriched in LREE, $Pb_c$, and U relative to zircon
residues (Fig. 3, Fig. 4, Fig. 5, and Table S2). LREE enrichment is apparent both in chondrite-
normalized REE spider diagrams and in LREE-I values. Sample's LREE-I and radiogenic to



Figure 4. AS3 U-Pb and trace element data for the 180 °C (left) and 210 °C (right) experiments. **(a)** LREE-I versus percent discordance. The horizontal solid line represents perfect concordance. The vertical dashed line depicts a LREE-I threshold value of 35 below which data is notably more discordant. **(b)** $^{206}Pb/^{238}U$ date plotted as a function of the radiogenic Pb* to common Pb ratio. Error bars for the percent discordant and $^{206}Pb/^{238}U$ data reflect propagated 2σ analytical uncertainties. Most error bars are smaller than data markers. **(c)** The radiogenic Pb* to common Pb ratio versus the LREE-I showing a positive correlation between the two variables.

common Pb ratio (Pb*/Pb$_c$) are clearly correlated. L1 leachates from the 210 °C dataset are

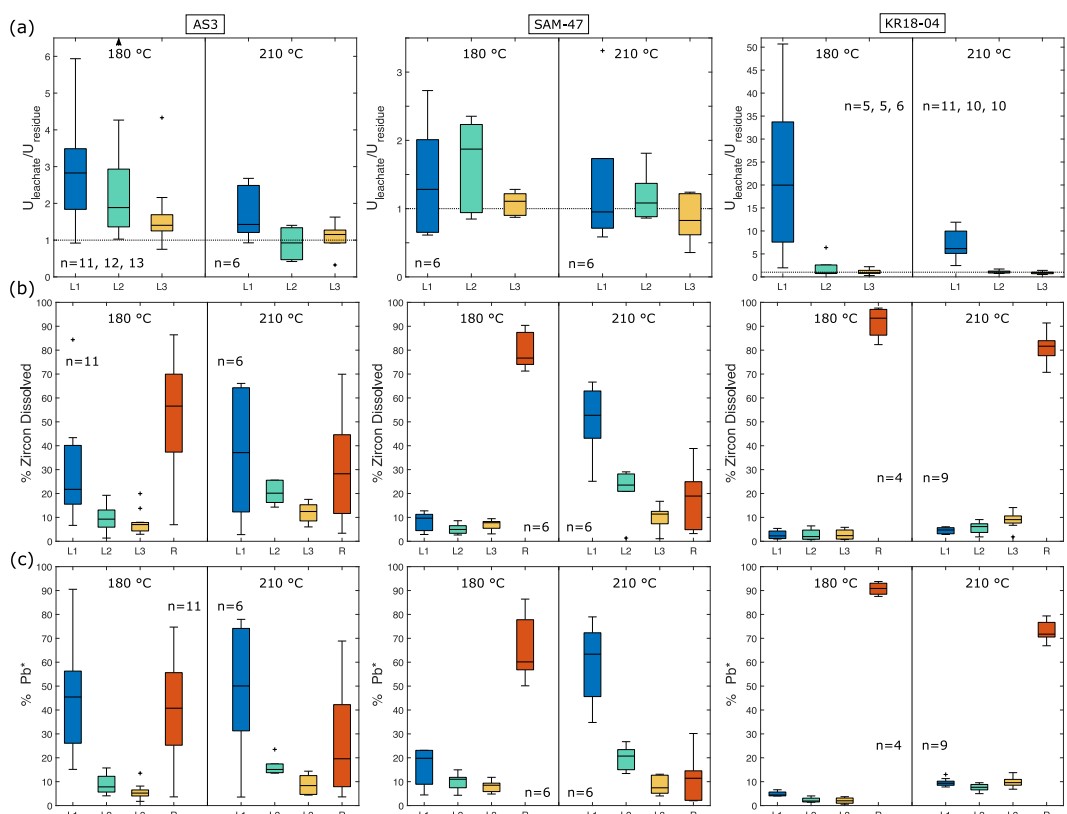

Figure 5. Box plot diagrams depicting geochemistry data for all step-leaching experiments. Each box shows the median value (black bar), the upper and lower quartiles (box), the minimum and maximum values (whiskers), and statistical outliers (plus marks) **(a)** Uranium concentration of leachates relative to that of their associated residue. **(b)** Percent zircon dissolved per leaching step based on measured Zr abundances. **(c)** Percent of radiogenic Pb measured per leaching step.

similarly enriched in LREE, $Pb_c$, and U relative to zircon residues, whereas 210 °C L2 and L3
leachates have compositions comparable to residues.
In the majority of samples, most dissolution occurred in the first leaching step with
progressively smaller volumes dissolved in L2 and L3. The median fraction of the zircon mass
remaining after L3 for the 180 °C experiment is ~55%, whereas for the 210 °C experiment it is
~30%. Percent Pb* mirrors results for percent zircon dissolved in both experiments.
**3.2 SAM-47**
**3.2.1 Geologic setting and sample description**
SAM-47 is an Archean (~3.32 - 3.29 Ga) granodiorite from the Corunna Downs Granitic Complex
of the Emu Pools Supersuite in the eastern Pilbara Craton (-21°24'29.01", 119°46'21.03")





(Barley and Pickard, 1999; Smithies et al., 2003; Van Kranendonk et al., 2007). The tectonic
significance of the dome and keel structures of the eastern Pilbara Craton are a matter of
debate, and the region has experienced a multi-phase deformational history (Kloppenburg et
al., 2001; MacLennan, 2019; Moore and Webb, 2013). ID-TIMS U-Pb cooling ages for apatite
from the Corunna Downs Granitic Complex are ~3.3 Ga which are similar to Ar-Ar ages reported
by Kloppenburg (2003). The similarity between the U-Pb and Ar-Ar data suggest rapid cooling
through ~460°C following intrusion of the granitoid (MacLennan, 2019). Zircon (U-Th)/He dates
for the Owen's Gully diorite from the Mount Edgar Granitic Complex north of the Corunna
Downs range from 677.5 ± 36.3 to 815.5 ± 44.6 Ma in age, suggesting that the eastern craton
reached near-surface thermal conditions where radiation damage can accumulate in zircon
sometime in the Neoproterozoic (Magee et al., 2017). Low-temperature thermochronology
data from elsewhere in the Pilbara craton (the northern, central, and western blocks) suggest
that the onset of widespread cooling related to basin-development and unroofing varied
regionally starting sometime between ~600 and 300 Ma (Morón et al., 2020). Zircon grains
separated from SAM-47 are euhedral, brown, and translucent. Crystals display fine-scale
concentric growth zones (McKanna et al., 2023). Rims are enriched in actinides and radiation
damage relative to cores. Raman data suggest that grains have accumulated intermediate-to-
high radiation damage densities with equivalent alpha doses ranging from $6\times10^{17}$ to $2\times10^{18}$ α/g
(McKanna et al., 2023). There is no previous U-Pb geochronology for zircon from this sample,
however, Pb loss is common in similarly aged zircon from the Pilbara craton (MacLennan, 2019).
**3.2.2 U-Pb and trace element results**
ID-TIMS U-Pb results for SAM-47 samples are presented in Fig. 2, Fig. 6, and Table S3. L1
leachates from both sample sets are normally discordant resulting in $^{206}Pb/^{238}U$ dates that are
>800 Ma. younger than zircon residues, indicating the dissolution of domains strongly affected
by Pb-loss. Although systematically older than L1 leachates, L2 and L3 leachates from the 180 °C
experiment are normally discordant. Two residues from the 180 °C dataset are concordant and
have a weighted-mean $^{206}Pb/^{238}U$ age of 3319.5 ± 1.4 Ma (MSWD = 1.7). The remaining four
residues are normally discordant. In contrast to the 180 °C experiment, many of the L2 and L3
leachates from the 210 °C experiment are concordant, and the few normally discordant
analyses closely approach Concordia. All 210 °C residues overlap or closely hug Concordia.
Three of the concordant residues yield a weighted-mean $^{206}Pb/^{238}U$ age of 3316.1 ± 1.6 Ma
(MSWD = 1.0) that is slightly younger than the 180 °C dataset. Upper intercept ages for 180 °C
residues, 210 °C residues, as well as 210 °C L2 and L3 leachates all agree within uncertainty and
produce robust MSWDs as shown in Fig. 6. The most precise upper and lower intercept ages are
3321.23 +0.78/-0.71 Ma and 751 ± 140 Ma, respectively.
Most $^{207}Pb/^{206}Pb$ dates for L2, L3, and R samples from the 210 °C experiment agree within
uncertainty (Fig. 2). The 210 °C residues yield a weighted-mean $^{207}Pb/^{206}Pb$ age of 3321.75 ±
0.63 Ma (MSWD = 0.83) which agrees well with our upper intercept ages. In contrast,
$^{207}Pb/^{206}Pb$ dates from the 180 °C dataset are notably younger, indicating the dissolution of
domains affected by a Pb loss event that occurred in the distant past. The two concordant 180

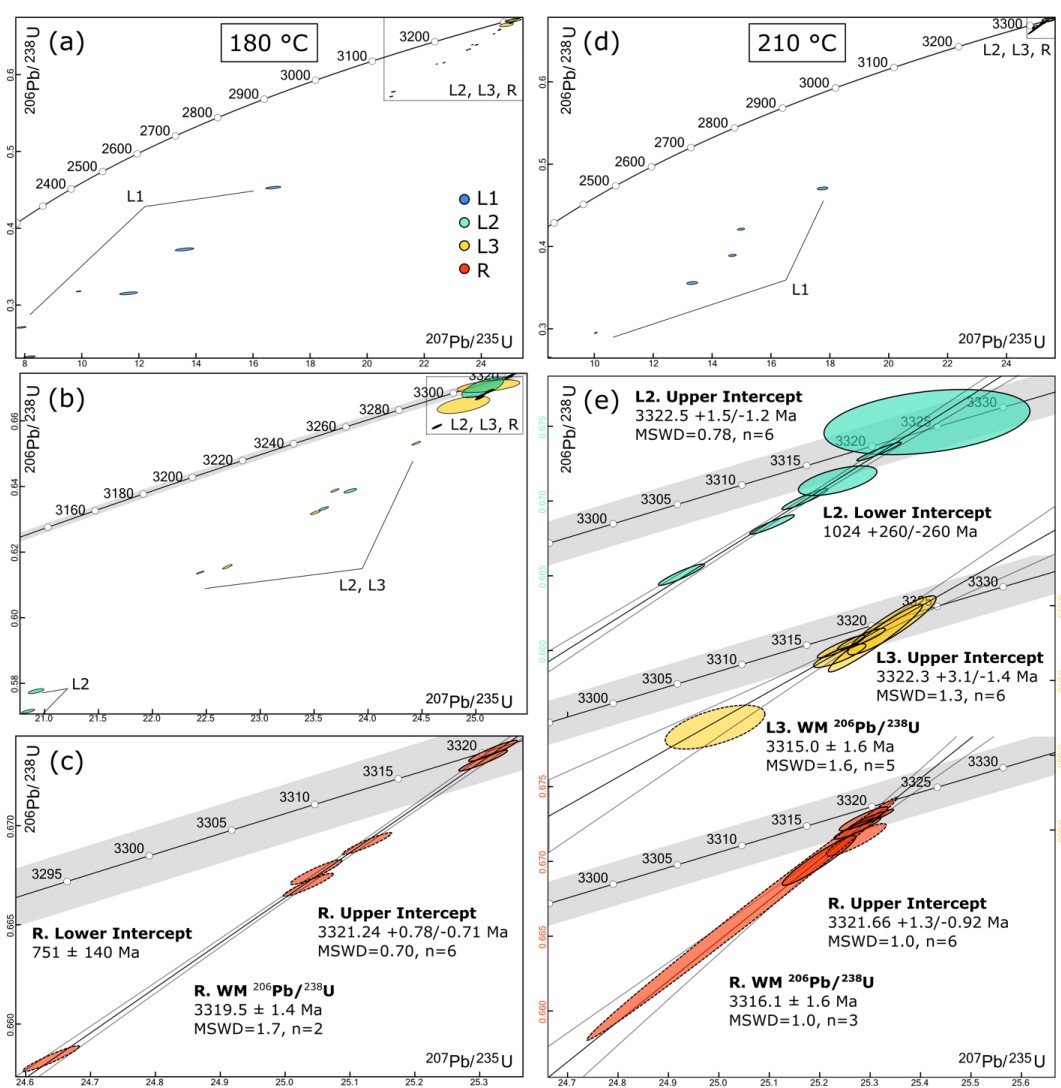

Figure 6. U-Pb concordia diagrams for the SAM-47 180 °C (left) and 210 °C (right) step-leaching experiments. **(a)** All data for the 180°C experiment. **(b)** Close up of the L2, L3, and R 180 °C dataset. **(c)** Close up of the 180 °C residue data. **(d)** All data for the 210°C experiment. **(e)** Stacked plot showing the L2, L3, and R 210 °C datasets. All ellipses reflect 2σ analytical uncertainties. Dashed ellipses are excluded from weighted mean calculations.

°C residue analyses yield a weighted-mean $^{207}Pb/^{206}Pb$ age of 3320.90 ± 0.87 (MSWD = 0.050) in
agreement with the 210 °C results.
L1, L2, and some L3 leachates from the 180 °C experiment are enriched in LREE and $Pb_c$ relative
to zircon residues (Fig. 7, Fig. 8, and Table S4). Some – but not all – leachates from the 180 °C
dataset are also marginally enriched in U relative to residues (Fig. 5). The composition of a
subset of 180 °C L3 leachates closely approximates that of residues. LREE and $Pb_c$ enrichment is

Figure 7. Chondrite-normalized REE spider diagrams for the 180 °C (left) and 210 °C (right) SAM-47 experiments comparing results for leachates and residues.

also evident in L1 and some L2 leachates from the 210 °C dataset, whereas most L3 leachates
have compositions comparable to residues. Although a few leachates from the 210 °C
experiment are relatively enriched in U, many have U compositions similar to residues.



GEOCHRONOLOGY Open Access
Discussions
EGU

Figure 8. SAM-47 U-Pb and trace element data for the 180 °C (left) and 210 °C (right) experiments. **(a)** LREE-I versus percent discordance. The horizontal solid line represents perfect concordance. The vertical dashed line depicts a LREE-I threshold value of 20 below which data is notably more discordant. **(b)** $^{206}$Pb/$^{238}$U date plotted as a function of the radiogenic Pb* to common Pb ratio. Error bars for the percent discordant and $^{206}$Pb/$^{238}$U data reflect propagated 2σ analytical uncertainties. Most error bars are smaller than data markers. **(c)** The radiogenic Pb* to common Pb ratio versus the LREE-I showing a positive correlation between the two variables.



### 3.3 KR18-04

#### 3.3.1 Geologic setting and sample description

KR18-04 zircons come from a Neoproterozoic rhyolite body associated with the glaciolacustrine Konnarock Formation in the Blue Ridge Mountains of Virginia, USA (MacLennan et al., 2020) (36°41'47.95", 81°24'22.08"). The Konnarock Formation is part of a structurally continuous sedimentary sequence deposited in a continental rift environment (Merschat et al., 2014). This sequence unconformably overlies gneisses that are related to the Mesoproterozoic Grenville orogeny. ID-TIMS U-Pb ages for zircon separated from KR18-04 were used to show that glacial sedimentation was occurring at tropical latitudes at ~751 Ma, 30 million years prior to the Sturtian Snowball Earth (MacLennan et al., 2020). The post-depositional history of the region is complex and poorly resolved (Roden, 1991). Zircon fission track dates ($T_c$ = ~205°C) from the Blue Ridge are variably reset by burial reheating and range in age from ~617 Ma to late Paleozoic dates (Naeser et al., 2016). Zircon (U-Th)/He dates ($T_c$ = ~180 °C for crystalline zircon) from the Blue Ridge are contemporaneous with the late-stages of the Alleghenian orogeny indicating that the zircon He chronometer was fully reset by burial reheating and records synorogenic exhumation (Basler et al., 2021).

The KR18-04 rhyolite is crystal-rich with prominent, dominantly euhedral K-feldspar and quartz phenocrysts (MacLennan et al., 2020). Zircon grains separated from KR18-04 are euhedral, pink-orange, transparent, and minimally included. Grains exhibit concentric zoning in cathodoluminescent images with some faint, broad growth zones (McKanna et al., 2023). Raman data suggest that grains have accumulated low-to-intermediate radiation damage densities with equivalent alpha doses ranging from $5 \times 10^{16}$ to $2 \times 10^{17}$ α/g (McKanna et al., 2023).

#### 3.3.2 Previous geochronology

Twelve single-crystal zircon ID-TIMS U-Pb analyses for KR18-04 are presented by MacLennan et al., (2020). Zircon were initially chemically abraded at 185 °C for 12 h. However, since many of these analyses retained significant Pb loss, the intensity of chemical abrasion was increased to 210 °C for up to 14 h for the remaining samples. The twelve reported $^{206}$Pb/$^{238}$U dates – which combine both leaching conditions – range from 753.08 ± 0.33 to 741.21 ± 0.35 Ma. The reported data are statistically over-dispersed for a single population. The authors attribute the spread in ages along Concordia and the one discordant analysis to residual Pb loss (their Fig. S10). The reported eruption age for the sample derived from the eight oldest analyses and determined using a Bayesian Markov Chain Monte Carlo technique is 752.60 +0.12/-0.65 Ma.

#### 3.3.3 U-Pb and trace element results

ID-TIMS U-Pb results for KR18-04 samples are presented in Fig. 9 and Table S5. L1 leachates from both sample sets are affected by Pb loss; L1 leachates with Pb*/Pb$_c$ ratios >1 are normally discordant and >150 Ma younger than zircon residues. The lower intercept value for L1 leachates from the 210 °C experiment suggests zero-age Pb-loss. L2 leachates from both



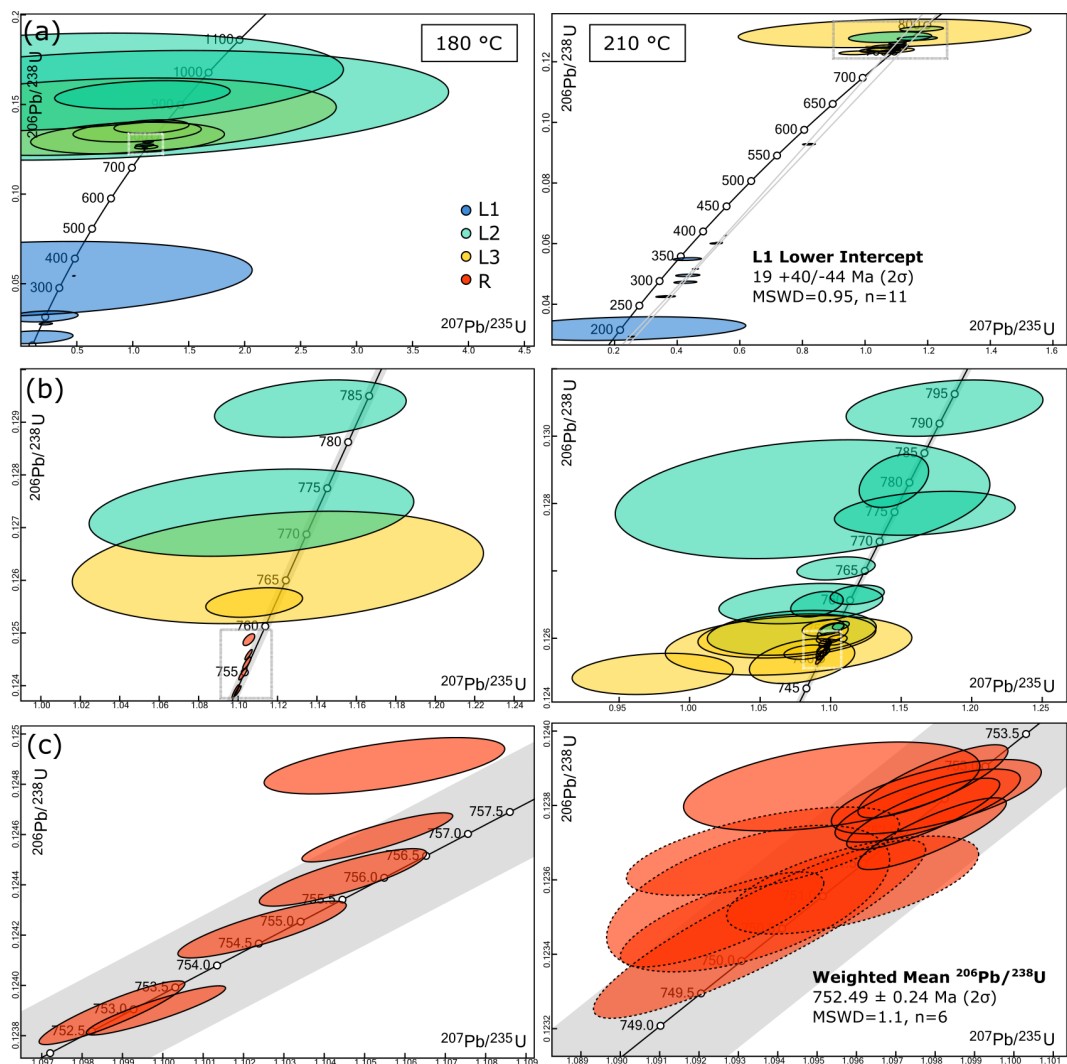

Figure 9. U-Pb concordia diagrams for the KR18-04 180 °C (left) and 210 °C (right) step-leaching experiments. **(a)** All data are depicted. **(b)** Close up of L2, L3, and R data excluding leachates with Pb*/Pb$_c$ values < 1. **(c)** Close up of zircon residues. The weighted mean $^{206}$Pb/$^{238}$U date reported for the 210 °C experiment includes residue data with solid ellipse borders. Ellipses with dashed borders were excluded due to low-quality U measurements. All ellipses reflect 2σ analytical uncertainties.

from the 210 °C experiment form a tight cluster with a weighted mean $^{206}$Pb/$^{238}$U age of 752.49 ± 0.24 Ma (MSWD = 1.1, n=6) in agreement with previous geochronology (MacLennan et al., 2020). This weighted-mean age includes analyses measured on the ATONA which produced more precise U measurements for these low-U zircon. The two 210 °C zircon aliquots that followed slightly different step-leaching protocols as outlined in Methods generated equivalent results.



Figure 10. Chondrite-normalized REE spider diagrams for the 180 °C (left) and 210 °C (right) KR18-04 experiments comparing results for leachates and residues.

in U, while L2 leachates are mildly enriched in U. L3 leachates have U compositions that are similar to residues (Fig. 5). L1 and some L2 leachates from the 210 °C experiment are enriched in LREEs relative to zircon residues, whereas L3 leachates have REE compositions similar to residues. All 210 °C leachates have low $Pb^*/Pb_c$ ratios compared to residues. L1 leachates are



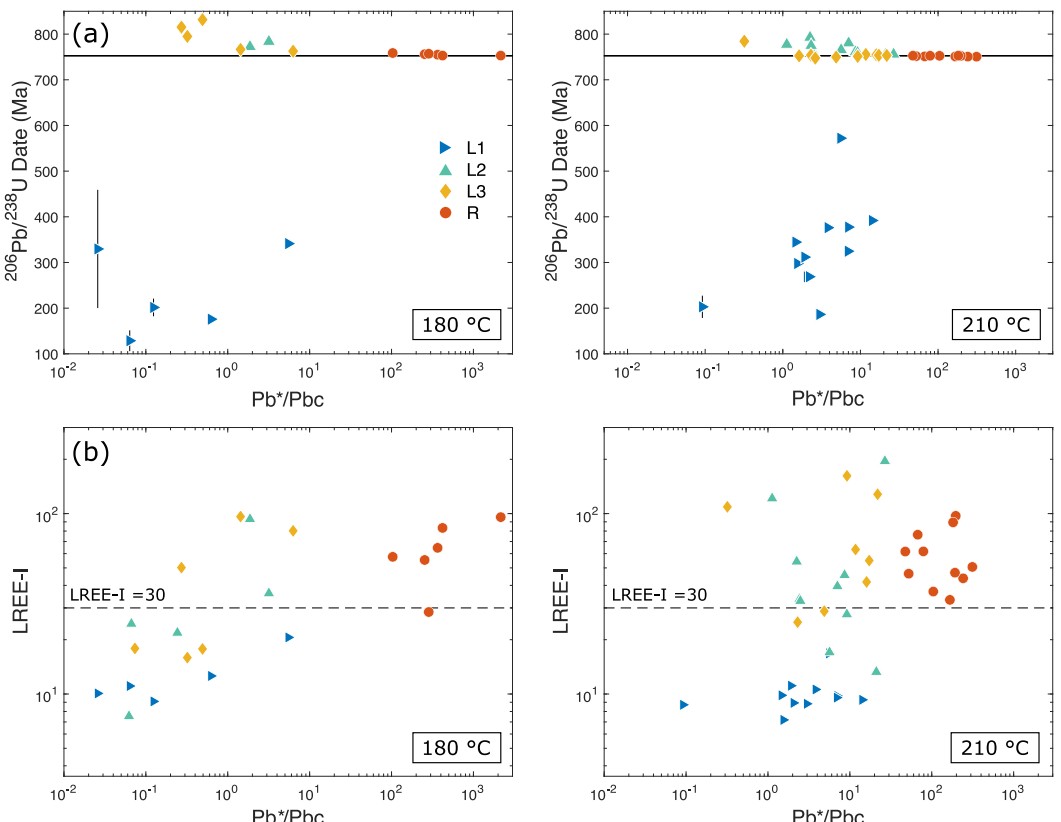

Figure 11. KR18-04 U-Pb and trace element data for the 180 °C (left) and 210 °C (right) experiments. **(a)** $^{206}$Pb/$^{238}$U date plotted as a function of the radiogenic Pb* to common Pb ratio. Error bars for the percent discordant and $^{206}$Pb/$^{238}$U data reflect propagated 2σ analytical uncertainties. Most error bars are smaller than data markers. **(b)** The radiogenic Pb* to common Pb ratio versus the LREE-I.


Only ~10 to 15 % of KR18-04 zircon mass dissolved during step-leaching at 180 °C, leaving ~85 to 95 % of the zircon residue available for final isotopic analysis. At 210 °C, ~10 °C to 30 % of the zircon dissolved during step-leaching, resulting in residue volumes of 70 to 90 %. Percent Pb* mirrors results for percent zircon dissolved in both experiments.


## 4. Discussion


### 4.1 Reverse discordance


Reverse discordance and concordant analyses that are older than the samples' interpreted crystallization ages are common in the AS3 and KR18-04 datasets but absent in SAM-47. Concordant analyses that are "too old" can result from either minor U loss or Pb* gain, causing datasets to lie along a discordia line that overlies the Concordia curve; for brevity, we will also refer to these analyses as "reversely discordant." Reverse discordance is most common in L2 and L3 leachates, however, a subset of residues from the AS3 and KR18-04 180 °C experiments




are also reversely discordant. Three L2 leachates for the Hadean zircon analyzed by Keller et al.,
(2019) are similarly reversely discordant.
Reverse discordance in zircon stepwise dissolution experiments is generally attributed to
leaching-induced experimental artefacts. Early step-leaching efforts yielded U-Pb isotopic
variations that swung wildly between normally and reversely discordant from step-to-step
(Todt and Büsch, 1981). Mattinson (1994, 2011) later attributed this effect to the authors'
specific dissolution and spiking method which caused U and Pb to fractionate between
supernate and U-bearing fluoride precipitates. However, later step-leaching experiments using
different experimental procedures also exhibited reverse discordance in early leaching steps
(Chen et al., 2001; Mattinson, 2005, 2011). Mattinson (2005, 2011) charged that early leaching
steps must reflect a mixture of U and Pb from the higher-U dissolved zircon volume plus excess
Pb* leached from the lower-U intact zircon residue. Mattinson (2005, 2011) further
demonstrated that annealing samples at temperatures between 800 - 1100 °C prior to chemical
abrasion helped to minimize – but not eliminate – leaching-induced artefacts.
Reverse discordance is observed naturally in some untreated zircon (Kusiak et al., 2015;
Wiemer et al., 2017; Williams et al., 1984). In such cases, reverse discordance is generally
attributed to either the internal redistribution of Pb within a crystal or by external factors such
as alteration by hydrothermal fluids (Mattinson et al., 1996). Alpha recoil can displace Pb* from
the position of its parent radioisotope by ~30 nm (Ewing et al., 2003; Weber, 1990, 1993). In
crystals with fine-scale growth zoning, Pb* produced by a U atom within a high-U zone can be
implanted into a nearby low-U zone producing a localized occurrence of excess Pb* in the low-
U zone (Mattinson et al., 1996). Further, ion imaging and atom probe tomography studies of
zircon support the case for nano-to-micro scale Pb redistribution under elevated temperatures
and pressures (Kusiak et al., 2015; Peterman et al., 2019, 2021; Reddy et al., 2016). These
studies show that unsupported Pb* often forms clusters that are not spatially associated with
parent radionuclide growth patterns. However, the exact mechanisms by which Pb* migrates
through the zircon structure are poorly understood.
Notably, our SAM-47 zircon does not exhibit reverse discordance suggesting that only some
samples are predisposed to leaching-induced artifacts or leaching-exposed natural U-Pb
fractionation. A zircon's U-Pb systematics as revealed by stepwise dissolution must therefore
reflect its unique compositional characteristics such as the length-scale and magnitude of
radionuclide zonation, the extent of Pb loss, or the sample's geological history. AS3 is
hydrothermally altered, so a component of the reverse discordance observed could potentially
reflect the redistribution of Pb isotopes during hydrothermal alteration (Takehara et al., 2018).
Why KR18-04 zircon is susceptible to reverse discordance is less clear. Grains appear unaltered
and most compositional zones are broad; however, some grains do have thin, high-U zones that
could contribute to the internal redistribution of Pb* (McKanna et al., 2023 their Fig. 4 & 15a).
Zircon fission track and (U-Th)/He data from Blue Ridge indicate that the region was thermally
affected by burial reheating during the late-Paleozoic Alleghenian Orogeny (Naeser et al., 2016;
Roden, 1991). Still, there is no evidence that KR18-04 has experienced an extreme high-



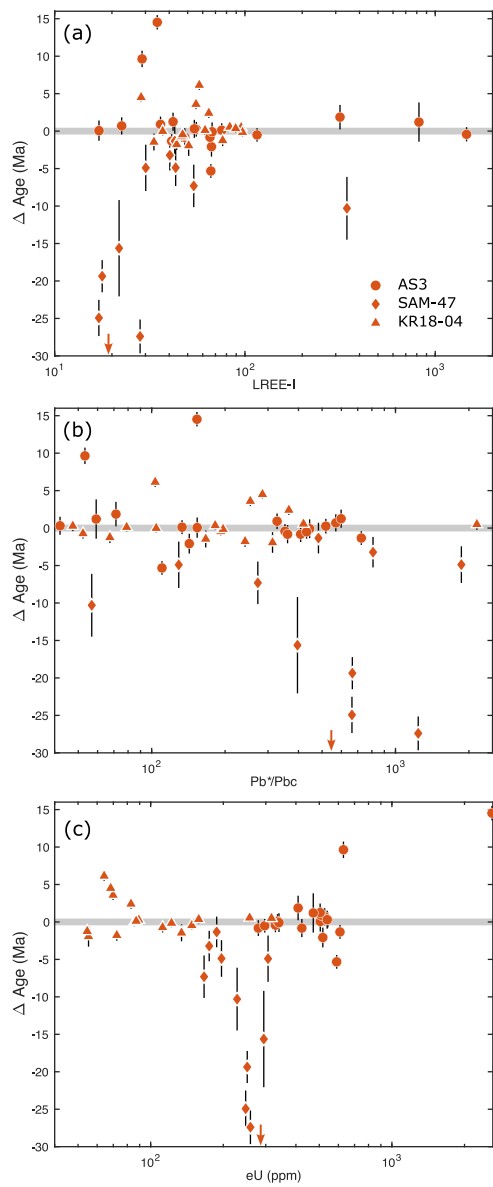

Figure 12. Trace element and Pb isotopic composition of zircon residues plotted against Δ Age as described in text. The gray bar at Δ Age = 0 Ma marks the accepted crystallization age for each zircon sample. The data portrayed is inclusive of residues from both the 180 °C and 210 °C experiments. **(a)** LREE-I versus the Δ Age. **(b)** Pb*/Pbc versus the Δ Age. **(c)** eU versus Δ Age. The arrow in each plot marks the placement of a datapoint from the SAM-47 dataset that plots at Δ Age = -60 Ma.

temperature deformation event. SAM-47 may lack leaching-induced reverse discordance simply
because Pb loss in the sample is so pervasive.



Regardless of the underpinning causes of reverse discordance, this work and that of Mattinson
(2005, 2011) demonstrate that increasing the leaching duration and/or temperature helps to
eliminate zircon domains affected by open system behavior. These results also highlight that
under-leaching samples can produce over-dispersed U-Pb datasets fraught with geologically
meaningless analyses. Without the additional context that the 210 °C experiment provides, a
researcher could easily interpret the older concordant dates from the 180 °C KR18-04 dataset,
for example, as inheritance or prolonged magmatic residence. We stress, however, that our
step-wise experiments are under-leached compared to the normal 12 h leaching step used in
most labs (see Section 4.3).
**4.2 The strengths and limitations of geochemical tools for identifying open-system behavior**
Common Pb and LREEs are incompatible in zircon. Mineral and melt inclusions and
hydrothermally altered or metamict zones, however, tend to be enriched in LREEs and common
Pb (Bell et al., 2016, 2019). Consequently, geochemical indicators such as a sample's LREE-index
(LREE-I = [Dy]/[Nd] + [Dy]/[Sm]) and $Pb^*/Pb_c$ (provided demonstrably low laboratory blanks) are
useful tools for identifying contamination, hydrothermal alteration, and metamictization.
Indeed, our data show that the two variables are generally positively correlated (Fig. 4C, 8C,
and 11B). Another important geochemical indicator is U concentration – or effective U
concentration (eU = U + 0.235 × Th) – which is a measure of the relative radiation damage in a
sample; zircon crystals or leachates from the same sample with higher eU have more radiation
damage than crystals or leachates with lower eU.
These three geochemical indicators are useful tools for evaluating zircon dissolution. In the 180
°C experiments, L1, L2, and some L3 leachates are enriched in LREE, $Pb_c$, and U relative to zircon
residues. Whereas in the 210 °C experiments, L1 and some L2 leachates are enriched in the
three variables, however, some L2 and most L3 leachates have compositions similar to residues.
Micro-X-ray computed tomography data presented by McKanna et al., (2023) for AS3 and SAM-
47 zircon show that acid readily accesses crystal cores via fractures to dissolve mineral and melt
inclusions and strongly metamict zones during L1 at 180 °C and 210 °C. As such, we interpret
the LREE, $Pb_c$, and U enrichment in L1 leachates to reflect the dissolution of inclusions,
metamict material, and – in the case of AS3 – hydrothermally altered zones. We attribute LREE,
$Pb_c$, and U enrichments in later leaching steps to the continued dissolution of soluble radiation-
damaged or altered domains. KR18-04 zircon grains are more crystalline and typically lack
fractures. Consequently, acid only accesses the cores of some grains, and some inclusions
armored by highly crystalline material appear to survive twelve hours of chemical abrasion at
180 °C or 210 °C (McKanna et al., 2023). Consequently, LREE and $Pb_c$ enrichment in L2 and L3
leaching steps could reflect later-stage dissolution of inclusions as well as the continued
dissolution of radiation-damaged or altered domains.
Comparing leachate and residue chemistry is extremely effective at illuminating the progress of
zircon dissolution. However, stepwise chemical abrasion is a time- and labor-intensive process.
The overwhelming majority of zircon ID-TIMS U-Pb studies perform single-step chemical



abrasion and discard the leachate. Only the residue is characterized. In an ideal scenario,
geochemical indicators such as those described here could be used to support the inclusion or
exclusion of anomalously young (or old) analyses in geochronological interpretations. Fig. 12
shows ΔAge (Ma) of residues plotted as a function of a grain's LREE-I, Pb*/Pb$_c$, or eU. ΔAge is
calculated as the difference between a residue's measured $^{206}$Pb/$^{238}$U date and each sample's
accepted crystallization age. Negative values for ΔAge reflect Pb loss, while positive values
indicate reverse discordance.
Unfortunately, there is no clear correlation between either of the three geochemical indicators
and ΔAge in the samples analyzed. Instead, the data suggest that relative enrichments in LREE,
Pb$_c$, and U in residues are not reliable indicators of residual open system behavior. We
speculate that the residual zircon affected by open-system behavior is likely volumetrically
small compared to the volume of the residual closed-system zircon. Thus, the geochemical
signature of the open-system behavior is likely masked by the bulk chemistry of the closed-
system residue. Relative enrichments in LREE, Pb$_c$, and U in residues are likely useful
geochemical indicators only if the residual open-system material is proportionally large.
**4.3 Leaching temperature and one-step versus stepwise chemical abrasion**
Stepwise dissolution at 210 °C out-performed stepwise dissolution at 180 °C for all three zircon
samples and produced more consistent, concordant datasets. Leaching at 210 °C dissolved
zircon material affected by open-system behavior earlier in the leaching process minimizing the
frequency and magnitude of normal and reverse discordance compared to the 180 °C
experiments (Figures 1, 2, 6, 7, and 10). The efficacy of the hotter leaching temperature is also
evident in zircon geochemistry; leaching at 210 °C more efficiently removed zircon material
enriched in U, LREE, and Pb$_c$.
Notably, U-Pb results for AS3 and KR18-04 residues treated by stepwise dissolution at 180 °C
are markedly worse than previous studies (MacLennan et al., 2020; Schoene et al., 2006).
Chemical abrasion of AS3 zircon for 12 to 14 h at 180 °C by Schoene et al., (2006) produced
concordant, statistically significant weighted mean U-Pb ages without signs of residual Pb loss
or reverse discordance. Those authors used intense frantzing to target unincluded, diamagnetic
zircon, whereas this study included altered grains. While some KR18-04 grains treated at 185 °C
for 12 h by (MacLennan et al., 2020) exhibited Pb loss, none of their chemically abraded
residues were found to be anomalously old or reversely discordant.
These apparent discrepancies beg the question: is a single 12 h leaching step is equivalent to
stepwise dissolution in three 4 h leaching steps? PTFE has a low thermal conductivity making it
an effective insulator. To evaluate how temperature in the PTFE-lined Parr pressure dissolution
vessel changes with time, a small hole was drilled through the top of an old PTFE liner. The
pressure vessel was assembled as normal minus the rupture and corrosion disks. A type-K
thermocouple with an insulated wire was threaded through the top of the pressure vessel and
into the center of the PTFE liner. The pressure vessel was then placed in a box furnace at 180 °C

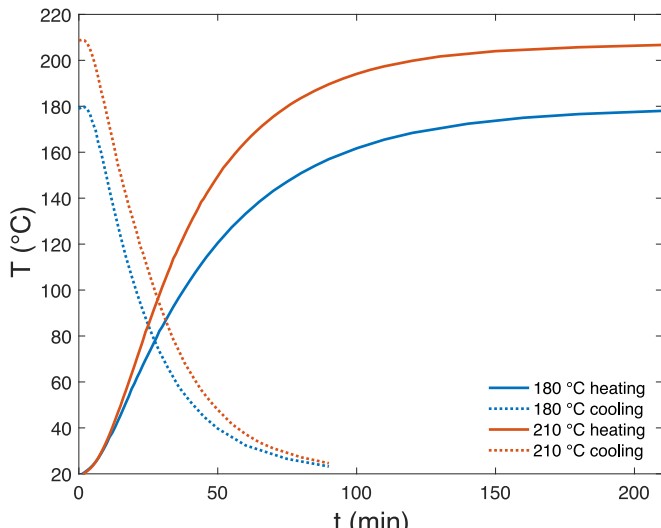

Figure 13. Temperature of the PTFE-lined pressure dissolution vessel plotted as a function of time. A fan was used to speed up cooling.

or 210 °C. Temperature was monitored using a Perfect Prime thermocouple thermometer until
the temperature in the liner reached equilibrium with the box furnace. The pressure vessel was
then removed from the furnace and placed in front of a fan, and temperature was recorded as
the pressure vessel cooled to near room temperature.
Results indicate that PTFE is indeed a very effective insulator; the interior of the pressure vessel
heats and cools slowly (Fig. 13). It takes 90 to 95 minutes for the interior of the pressure vessel
to reach within 20 °C of the target temperature and an additional 30 to 35 minutes to reach
within 10 °C of the target temperature. The pressure vessel takes ~90 minutes to cool to room
temperature once removed from the oven.
Given the heating ramp up and cool down times for the PTFE-lined pressure dissolution vessel,
samples spend only ~2 h of the 4 h leaching step within 10 °C of the target temperature. As
such, a sample leached for 12 h in three consecutive 4 h steps spends ~6 h within 10 °C of the
target temperature. Conversely, a sample leached in a single 12 h step spends ~10 h within 10
°C of the target leaching temperature – ~4 h longer than the step-leached sample. Volume loss
estimates for KR18-04 further support this conclusion; estimated volume losses for stepwise
chemical abrasion (Figure 5) are generally lower than the estimated volume losses presented by
McKanna et al., (2023; their Fig. 18) for zircon from the same sample aliquot that were
chemically abraded for a single 12-h step.
On the basis of the 10 °C threshold, we estimate that our dated residues have been leached for
a duration equivalent to a single ~8 h leaching step. Given our U-Pb results, we conclude that
zircon chemically abraded at 180 °C for a single 8 h step are "undercooked" and will likely



produce data affected by residual Pb loss and/or leaching-induced artifacts. Zircon samples
chemically abraded at 210 °C for a single 8-h leach are more likely to produce geologically
meaningful results. Unfortunately, we therefore cannot comment on the efficacy of the
routinely practiced 12-hour leaching at 180 ˚C used in many labs, except to say it is likely more
effective than the results for residues we present here.
**564  4.4 The relationships between alpha dose, Pb loss, and zircon dissolution: Moving toward a**
**565  more predictable model for chemical abrasion**
Zircon is an outstanding chronometer because radiogenic Pb is immobile in well crystalline
zircon (Cherniak et al., 2009; Cherniak and Watson, 2000). Establishing the alpha dose at which
radiogenic Pb can mobilize within the zircon structure would help make Pb loss more
predictable. We calculate three different time-integrated alpha doses for each sample using the
radionuclide concentrations determined for leachates and residues (Table 1). "Total" alpha
dose assumes a damage accumulation interval equivalent to a sample's crystallization age. This
calculation ignores the possibility of radiation damage annealing. "Present day" alpha dose
estimates attempt to take geological annealing into account. Radiation damage anneals at
temperatures above ~200 to 300°C on geological timescales (Bernet, 2009; Yamada et al.,
2007). The closure temperature for the (U-Th)/He system in crystalline zircon is ~180 °C
(Guenthner et al., 2013; Reiners et al., 2004). As such, we use published zircon (U-Th)/He dates
or thermal histories derived from zircon (U-Th)/He datasets for the Minnesota River Valley
(Guenthner et al., 2013; McDannell et al., 2022), the Eastern Pilbara craton (Magee et al.,
2017), and the Virginia Blue Ridge (Basler et al., 2021) to estimate minimum damage
accumulation intervals for samples' "present day" alpha doses. Since zircon (U-Th)/He dates for
the Eastern Pilbara craton broadly overlap the lower-intercept U-Pb Concordia age for SAM-47,
we take the lower-intercept age as the damage accumulation interval. Chosen intervals for AS3,
SAM-47, and KR18-04 are 750 Ma, 751 Ma, and 298 Ma, respectively.
"Present day" alpha dose estimates can also be established independently using Raman
spectroscopy, since key bands in the zircon Raman spectrum broaden predictably with

Table 1. Alpha dose estimates.

| Sample | α dose (α/g)[1] | | | | | | $\alpha_r$ dose (α/g)[2] | |
|---|---|---|---|---|---|---|---|---|
| | Total | | Pb Loss | | Present Day | | Present Day | |
| | Min | Max | Min | Max | Min | Max | Min | Max |
| AS3 | $4\times10^{17}$ | $1\times10^{20}$ | $3\times10^{17}$ | $8\times10^{19}$ | $3\times10^{17}$ | $8\times10^{19}$ | $2\times10^{17}$ | $>1\times10^{19}$ |
| SAM-47 | $2\times10^{18}$ | $1\times10^{19}$ | $1\times10^{18}$ | $8\times10^{18}$ | $3\times10^{17}$ | $2\times10^{18}$ | $6\times10^{17}$ | $2\times10^{18}$ |
| KR18-04 | $1\times10^{17}$ | $1\times10^{19}$ | $4\times10^{16}$ | $4\times10^{18}$ | $4\times10^{16}$ | $4\times10^{18}$ | $5\times10^{16}$ | $7\times10^{17}$ |

[1]Calculated using measured U and Th concentrations and damage accumulation intervals as described in text.

[2]Raman-based alpha dose estimates reported by McKanna et al., (2023).

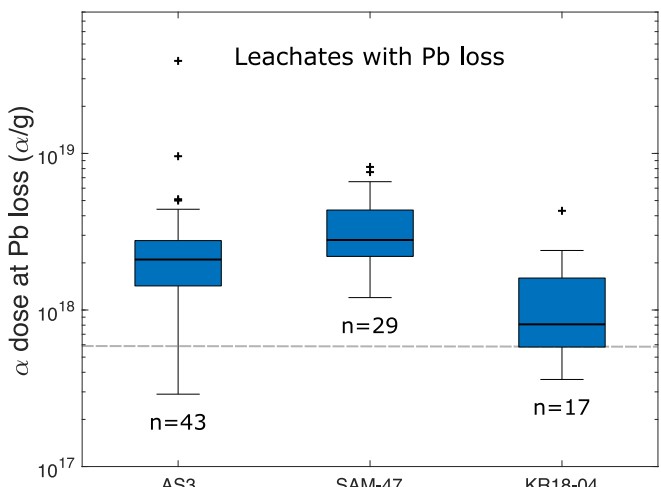

Figure 14. Box plot diagram showing alpha dose distribution for leachates (L1, L2, and L3) affected by Pb loss. Data include both the 180 °C and 210 °C experiments. The gray dashed line highlights our best estimate for the minimum alpha dose required for Pb loss to occur. Each box shows the median value (black bar), the upper and lower quartiles (box), the minimum and maximum values (whiskers), and statistical outliers (plus marks).

increasing alpha dose (Nasdala et al., 2001; Palenik et al., 2003). "Present day" alpha doses for
AS3 and SAM-47 closely match Raman-based alpha doses ($\alpha_r$) determined by (McKanna et al.,
2023) for zircon from the same sample aliquots (Table 1). "Present day" alpha doses for KR18-
04 have a similar lower bound but a higher upper bound compared to Raman estimates. Most
likely, Raman measurements failed to capture volumetrically small, higher-U domains such as
the thin concentric dissolution features evident in secondary electron images of KR18-04
residues (McKanna et al., 2023, their Fig. 15a-I reproduced here in Fig. 16b).
The final calculation estimates alpha dose at the time of Pb loss. Because AS3 and KR18-04
exhibit zero-age Pb-loss discords, "present day" and "Pb loss" alpha doses estimates are
equivalent. The Pb-loss discord for SAM-47, however, suggests that Pb loss occurred in the
distant geological past at or before 751 ± 140 Ma (Fig. 6C). Therefore, the maximum "Pb loss"
damage accumulation interval is the difference between the sample's upper and lower
intercept ages which equates to ~2571 Ma.
Fig. 14 shows the distribution of "Pb loss" alpha dose estimates for all leachates affected by Pb
loss. Despite vastly different geological settings and ranges in radiation damage densities,
leachates affected by Pb loss exhibit similar alpha dose distributions. The majority have alpha
doses that are $\geq 6 \times 10^{17}$ $\alpha$/g. We therefore establish this alpha dose as our best estimate for
the threshold above which Pb can mobilize within the zircon structure. Notably, this threshold
is somewhat lower than the alpha dose $- 1 \times 10^{18}$ $\alpha$/g – at which zircon material properties such
as density begin to change (Ewing et al., 2003; Nasdala et al., 2004). However, the $6 \times 10^{17}$ $\alpha$/g
threshold is similar to some estimates for the alpha dose at which helium diffusion kinetics

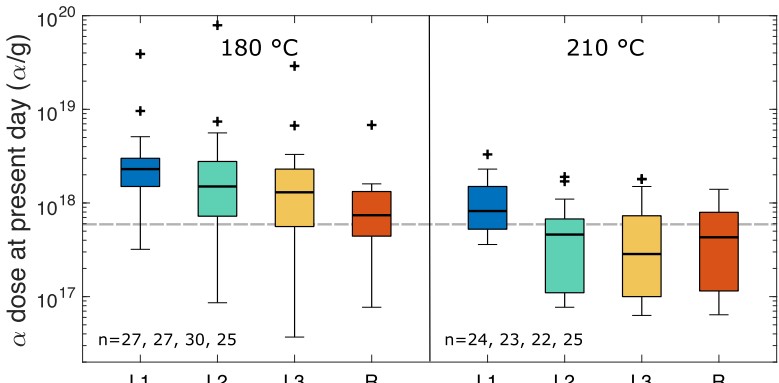

Figure 15. Box plot diagram showing present day alpha dose distributions at each step of zircon dissolution. Data includes all AS3, SAM-47 and KR18-04 leachates and residues. Alpha dose estimates reflect samples' present day radiation damage. The gray dashed line highlights our best estimate for the minimum alpha dose required for Pb loss to occur. Each box shows the median value (black bar), the upper and lower quartiles (box), the minimum and maximum values (whiskers), and statistical outliers (plus marks).

begin to increase causing the closure temperature for He in zircon to decrease (Anderson et al.,
2017, 2020). The mechanism that causes Pb loss – diffusion, leaching, or recrystallization – is
not clear.
For the best geochronological outcomes, chemical abrasion should target zircon material with
alpha doses ≥ $6 \times 10^{17}$ α/g. Fig. 15 shows "present day" alpha dose estimates for all leachates
and residues from the 180 °C and 210 °C experiments. In the 180 °C experiments, the median
alpha dose decreases with increasing leaching duration consistent with the expected effects of
radiation damage on zircon solubility. A majority of residues from the 180 °C experiments have
alpha doses > $6 \times 10^{17}$ α/g suggesting that residues may be affected by residual open system
behavior in agreement with our U-Pb isotopic results. Evidently, dissolving zircon with lower
alpha doses requires longer leaching durations at 180 °C than achieved in this study, which was
equivalent to a single 8-hour leach step. In contrast, the median alpha dose for residues as well
as L2 and L3 leachates from the 210 °C experiments have alpha doses below the established
threshold. Zircon material with alpha doses ≥ $6 \times 10^{17}$ α/g is thus readily dissolved at short
leaching durations at 210 °C.
Framing Pb loss and zircon dissolution in terms of alpha dose better allows the user to tailor
their chemical abrasion approach to a specific zircon dataset. Fig. 16 plots alpha dose as a
function of time for different U concentrations. As described above, different time intervals can
be selected for damage accumulation depending on the calculation's goal. This figure is a
simple visual representation that can help a researcher determine whether or not a sample is
likely to be affected by Pb loss. Chemical abrasion is a time-consuming method that is applied
to the majority of ID-TIMS U-Pb datasets, but it may be unnecessary for low-alpha dose,
inclusion-free zircon. Further, by estimating a sample's "present day" alpha dose distribution, a
user can better predict how readily a sample will dissolve. For example, if a sample has a lot of

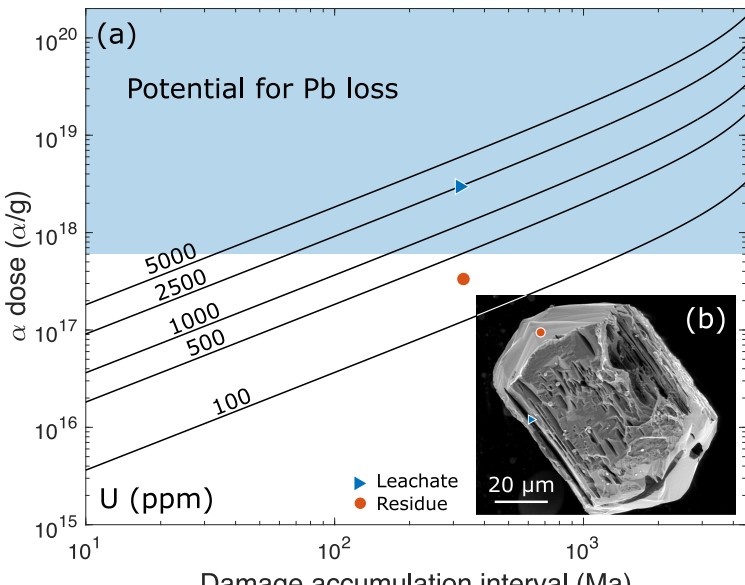

Figure 16. **(a)** Contour diagram showing alpha dose as a function of time for U concentrations ranging from 100 ppm to 5000 ppm. Calculations assume a fixed Th/U of 0.5 and no annealing. The shaded region highlights the alpha dose range in which Pb loss is most likely. **(b)** Secondary electron image of KR18-04 residue chemically abraded at 210 °C for 12 h from McKanna et al., (2023). The blue triangle marks a thin concentric zone that dissolved during chemical abrasion (leachate), while the red circle marks a portion of the zircon that remained intact (residue). Markers in b) correlate to markers in a) and illustrate how a grain with radionuclide zoning can have accumulated alpha doses above and below the threshold for Pb mobilization.

accumulated radiation damage like SAM-47, leaching longer than a single 8-h step at 210 °C will
likely leave little to no residue for isotopic analysis. Whereas, if a sample has a lower alpha dose
like KR18-04, a longer 210 °C leach is unlikely to have a significant effect.
As highlighted in Fig. 16b, perhaps the most persistent challenge when it comes to tailoring
chemical abrasion for a specific zircon dataset or even a specific zircon crystal is radionuclide
zoning. The spatial distribution of radionuclides and the magnitude of intracrystalline variations
in radiation damage strongly affect the mechanics of zircon dissolution (McKanna et al., 2023).
Radionuclide zoning explains the inconsistent dissolution behavior evidenced in Fig. 5. For
example, the percent zircon dissolved in each leaching step decreases from L1 to L3 for AS3
zircon, remains constant or decreases for SAM-47 zircon, and remains constant or increases for
KR18-04 zircon. This non-pattern occurs because the percent zircon dissolved is not only a
function of alpha dose, but also 1) the volumetric proportion of zircon with a given alpha dose,
and 2) which portions of a crystal are in contact with HF at any given time during the leaching
process. Building a comprehensive model for chemical abrasion will ultimately require both
geochemical and textural inputs.





## 5. Conclusions

Single-crystal stepwise dissolution experiments performed at 180 °C and 210°C provide new insights into the geochronological and geochemical effects of chemical abrasion on zircon datasets. Because of the insulating properties of the PTFE-lined pressure dissolution vessel, stepwise dissolution in three 4-h leaching steps is not equivalent to a 12-h single-step chemical abrasion, the method most commonly used by the zircon ID-TIMS U-Pb community. We estimate that our stepwise dissolution approach is roughly equivalent to 8-h single-step chemical abrasion. Stepwise dissolution at 180 °C produced over-dispersed U-Pb datasets affected by both residual Pb-loss and leaching-induced or leaching-exposed artefacts which present as reverse discordance. Without the context of the 210 °C results, reverse discordance in the 180 °C datasets could easily be mistaken for prolonged crystallization or inheritance and lead to spurious geological interpretations. Longer leaching durations are likely needed to produce robust geochronological datasets at 180 °C.

Stepwise dissolution at 210 °C outperformed the 180 °C experiments by all measures for the three zircon samples analyzed producing more reproducible, concordant results. Ultimately, how a zircon sample responds to any chemical abrasion protocol will be sample-dependent. However, our results suggest that 8-h single-step chemical abrasion at 210 °C may be effective at mitigating Pb loss and reverse discordance for a wide range of zircon samples. Further study of different zircon samples is needed. Our results, however, clearly demonstrate that leaching durations longer than an 8-h single step are required for chemical abrasion at 180 °C to be effective.

U concentration, $Pb^*/Pb_c$, and LREE enrichment are useful tools for tracking the dissolution of inclusions and radiation-damaged or altered material during stepwise dissolution. These geochemical indicators, however, are not effective at identifying residual Pb loss in the zircon residues analyzed.

We attempted to constrain the relationship between Pb loss and radiation damage by calculating an alpha dose for each leachate based on its measured radionuclide concentration and an estimated damage accumulation interval informed by the sample's geologic history. "Pb loss" alpha dose estimates suggest that Pb may mobilize within the zircon structure at alpha doses as low as $6 \times 10^{17}$ α/g. "Present day" alpha dose estimates indicate that many residues treated by stepwise dissolution at 180 °C have alpha doses above the $6 \times 10^{17}$ α/g threshold, and consequently, many 180 °C residues are affected by residual Pb loss. The majority of residues treated at 210 °C – and many L2 and L3 leachates – have "present day" alpha doses below this threshold. Grains expected to have accumulated alpha doses $< 6 \times 10^{17}$ α/g based on expected radionuclide concentrations and damage accumulation intervals are unlikely to be affected by Pb loss and may not require chemical abrasion. However, chemical abrasion may help improve the precision of U-Pb analyses even in low-damage grains by dissolving $Pb_c$-bearing inclusions. The effectiveness of any chemical abrasion protocol will ultimately be sample-dependent, since zircon dissolution depends not only on a grain's bulk chemistry, but also the spatial distribution and magnitude of intracrystalline variations in radiation damage.



**Data availability.** All data presented are included in this paper or the Supplement.

**Supplement.** The supplement related to this article is available online at:

**Author contributions.** AJM carried out the experiments and wrote the manuscript. All authors – AJM, DS, and BS – contributed to the experiment design and data reduction, interpretation, and presentation.

**Competing interests.** The contact author has declared that none of the authors has any competing interests.

**Disclaimer.** Publisher's note: Copernicus Publications remains neutral with regard to jurisdictional claims in published maps and institutional affiliations.

**Acknowledgements.** Thank you to Mami Takehara of the National Institute of Polar Research in Tokyo, Japan, for providing the hydrothermally altered AS3 zircon crystals used in this study.

**Financial support.** This work was supported by research funds provided by the Department of Geosciences at Princeton University granted to Alyssa J. McKanna as part of her Harry Hess Postdoctoral Fellowship.

**Review statement.**

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
