# Peer review of "Geochronological and Geochemical Effects of Zircon Chemical Abrasion: Insights from Single-2 Crystal Stepwise Dissolution Experiments"

_Geochronology, 2023_

## Referee Comment (RC2)

[referee-annotated manuscript omitted]

---

## Author Comment (AC1)

We thank the reviewers for their time and thoughtful commentary. Their feedback will undoubtedly strengthen the manuscript. Our responses to their suggestions are recorded below in blue.

**Reviewer #1**

The manuscript "Geochronological and Geochemical Effects of Zircon Chemical Abrasion: Insights from Single- Crystal Stepwise Dissolution Experiments" by McKanna and others is presenting step leaching U-Pb ID-TIMS data on three different zircon reference materials for two different temperatures and giving a recommendation for conditions to use by practitioners of chemical abrasion and an estimated minimum alpha dose at which Pb loss can occur. This study is a follow up from a previous study by the same authors on the same zircon samples that looked at structure of individual zircon grains of in response to chemical abrasion at different temperatures.

The conclusions from this study are largely confirming what has already been published about the chemical abrasion conditions and its consequences (Huyskens et al 2016, Widmann et al. 2019). All three studies concluded that leaching at 180C is not high enough for most samples to effectively remove all Pb loss from a sample and a higher temperature is needed in most cases. This study, however, is the only one that has detailed structural observations of zircons before and after the leaching procedure and it can tie them together with the U-Pb observations. This also leads to the possibility of estimated radiation damage of the leached zones, which could not be done in the two previous studies.

This is the main new angle of the study and these observations and discussions are unfortunately falling short in favour of documenting observations at length that have been made before. Thus, my recommendation is to focus a lot more on the new aspects of the study. To cut back on the repetitions reorganising of the results section would help. For example, for each sample early leachates are enriched in LREE. Thus keeping the standard format of first heading "geologic setting and sample description" with a sub heading of the different samples is advantageous. It also makes it easier to compare and contrast the different zircons analysed in this study.

We thank the reviewer for the suggestion. We will reorganize the results section in revisions to make it more concise and less repetitive.

The authors also looked for a tool to robustly identify zircons that have remaining Pb loss. Currently the identification of such analysis is very subjective and such a tool would make interpretations of a scattered dataset more robust. Unfortunately, neither of the indicators (U concentration, $Pb^*/Pb_c$, or LREE enrichment) are effective tools for this task.

Yes, the result is unfortunate, but important to demonstrate.

One of the recommendations of this paper is to look for the amount of radiation damage and tailor the leaching conditions this way. However, no information is given on how to determine the radiation damage prior to dissolution. In this study the radiation dose was calculated based on the analysis of U and Th in the leachate and residue of the zircons, which means after already performing the time-consuming analyses. In the prior study Raman was used to estimate the alpha dose for the zircons. Any method to determine the alpha dose would need to have a high spatial resolution for the entire volume of the zircon grain, since it was documented in the previous paper that these zones do exist also in the interior of some zircon grains.

Determining alpha dose prior to dissolution is tricky. The reviewer is correct that the distribution of radiation damage in zircon is also typically heterogeneous. Unfortunately, there isn't a method currently available to derive high spatial resolution alpha dose values for the entire volume of a the zircon grain. Micro X-ray computed tomography (µCT) can distinguish between crystalline and strongly-damaged zircon in three dimensions as demonstrated in our companion paper. However, it is not currently possible to derive alpha dose directly from µCT data. Radiation damage can be estimated using Raman spectroscopy, and Raman mapping can be used to characterize internal variations in radiation damage. Alternatively, one can use Raman spot analyses guided by CL-imagery to bracket minimum and maximum alpha doses (i.e., measure Raman spectra from CL bright areas and CL dark areas). We acknowledge that this can be a time-consuming process and requires access to a Raman system. Another approach would be to measure the U and Th contents of zircon in situ prior to dissolution using laser ablation ICPMS. Paired with an appropriate damage accumulation interval based on the sample's geological history, alpha dose can be calculated. Again, time- and instrument-intensive.

Figure 16 can provide a quick-and-dirty tool for estimating alpha dose in zircon crystals without a lot of information. Given a rough estimate for a sample's damage accumulation interval based on the sample's geology (such as an approximate crystallization age or cooling age from thermochronology data in the literature), one can calculate alpha dose for a range of possible U concentrations. We can add these points to the discussion in revisions. Ultimately, this study and other chemical abrasion studies suggest that leaching at 210 °C is effective for a wide-range of alpha doses.

Please provide all the calculated numbers that are used in the plots like alpha dose at Pb loss or LREE-I. – We can provide this information in a table in the Supplementary Materials.

A description is needed how the amount of dissolved material for the calculation of alpha dose was determined. – The only variables that the alpha dose calculation requires is the concentration of U and Th in the leachate/residue and time – i.e., the damage accumulation interval. The Th concentration was determined by ICPMS, and the U concentration is calculated from the Th concentration and the Th/U ratio determined by TIMS. The mass or volume of dissolved material is not needed.

Fig 12: could you distinguish between the different temperatures maybe using open vs filled symbols? – We can make the suggested change in revisions.

Fig 15: It looks like the samples leached at 210C overall had lower radiation damage compared to the ones used in the 180C experiment in this figure and this would need an explanation. My guess is that it has something to do with the fraction of material in each of the dissolution steps, but this is confusing at first. For me, the figures about the alpha dose are the most important in this study and the groupings in Fig 14 and 15 are so broad that they could be masking interesting details. A plot including alpha dose vs discordance for example on an individual analysis basis could be interesting.

The reviewer is correct that the apparent differences in alpha dose between the samples leached at 180 °C and 210 °C in Figure 15 reflect the fraction of material dissolved in each step. 180 °C leachates reflect the dissolution of small volumes of high-U zones. Whereas, leaching at 210 °C dissolves a larger volume of material including both high-U and medium-U zones, causing the average U concentration (and alpha dose) to be lower for 210 °C L1 leachates than for 180 °C L1 leachates. We will include discussion about this in revisions.

In general, we find considering the data in aggregate to be meaningful for deriving overarching trends. Alpha does vs. discordance is not a useful metric for evaluating the AS3 and the KR18-04 datasets, since the data spread along the concordia line (Fig. 1 and Fig. 9).

Line 31:" However, Ultimately,…" – We will fix this typo.

Line 49: "since the trajectory of Pb-loss follows Concordia" I think it should be the concordia. – We will fix this typo.

Lines 50-51: "the precision of 207Pb/235U dates is also lower than corresponding 238U/206Pb dates due to the shorter radioactive half-life of 235U and lower isotopic abundance (Corfu, 2013; Schoene, 2014)." The precision of the 207Pb/238U dates is lower not because of the shorter half-life. It is only lower due to the lower abundance, which in turn is due to the shorter half-life. – The reviewer is correct. We will add this to the revisions.

Line 59:" annealing zircon samples prior to leaching helps to minimize the unwanted isotopic fractionation effects that plagued earlier leaching attempts" The main improvement is reducing elemental fractionation in the leaching steps. – The reviewer is correct. We will add this to the revisions.

Line 191: "included and altered grains…" Grains with inclusions – We will correct this in revisions.

Line 210: "which had been redone as part of (Schoene et al., 2006)."   Brackets are in the wrong place – We will correct this in revisions.

Line 522: frantzing should not be a verb – We will change this in revisions.

---

## Author Comment (AC2)

We thank the reviewers for their time and thoughtful commentary. Their feedback will undoubtedly strengthen the manuscript. Our responses to their suggestions are recorded below in blue.

**Reviewer #2 – Fernando Corfu**

The paper reports the results of experiments on the chemical abrasion of zircon, using 3 samples of different age, and applying two different experimental protocols. The reported results include U-Pb isotopic ratios and ages, and abundances of a number of chemical element typical in zircon. The experiments evaluate the efficiency of the two different approaches in removing discordant zircon domains and isolating grains with closed isotopic systems.

The data will be interest to the geochronologists that use the U-Pb dating methods, especially the ID-TIMS community. It is not the first paper to report such experiments, but it adds some new perspectives, which will certainly be useful for a further advancement of the technique.

The paper is reasonably well prepared. There are some technical glitches, with figures inserted in the text and locally covering up portions of the text. I have put some suggestions and a number of comments directly in the annotated file.

We will address the technical glitches. Responses to specific line edits are listed below.

The tables need some work to make them more useful and accessible for the readers. (1) It would be practical to assemble them all as separate sheets in just one file. (2) It would be practical to list U abundances, since they are mentioned repeatedly in the text. At present one has to use the Th/U ratios from one table and combine them with the Th abundances in another table to get an idea of the U contents. (3) There is no explanation of what (ppt) stands for (part per trillion, or per ton?), and ppt of what? Some solution? Because of this enigma the listed numbers do not mean anything directly. Further back in the table there are then absolute abundances in ppm. Please, put those in the front, and explain all the terms used. (4) Please list the 206/204 ratios.

We will make the suggested changes to the tables. 1) We will list results for the different zircon samples in separate tabs in one U-Pb and one ICPMS file. 2) We will add U concentrations to the ICPMS file. 3) We will move the concentration of the element in zircon (ppm – parts per million) to the front of the ICPMS file. The concentration of the element in 1 mL solution (ppt – parts per trillion) is listed at the back of the ICPMS file. 4) We will add the $^{206}Pb/^{204}Pb$ ratios to the U-Pb file.

(5) The outcome of the experiments depends very strongly on the qualities and characteristics of the zircon grains used, but the tables provide no information in merit at all. One may perhaps try to link the individual data to the information in the previous associated paper by these authors. I highly recommend putting a characterization of each grain in the table. Geochronologists know that no two zircons are born alike, and they know that successful dating

is best done by a strict discrimination of the good from the bad. A lack of information on the tested grains strongly weakens the interpretations and lessons learned from the study.

We will add cathodoluminescence images (AS3 & KR18-04) and backscattered electron images (AS3 & SAM-47) of dated grains to the Supplementary Materials (SAM-47 grains were luminescent in CL). We feel this additional data will be more useful for evaluating the characteristics of dated grains than written descriptions in a table would. The general characteristics of the three zircon populations are discussed in detail in our GChrong companion paper: "Chemical abrasion: The mechanics of zircon dissolution."

The discussion comprises are section linking a-dosage and degree of discordance, and its implications for the CA-application. It is certainly true that a-dosage and the relative radioactive damage are important factors affecting discordance. But it is also very simplistic, and not realistic, to reduce the degree of discordance to a straight function of a-dosage. Clearly, the textural factors, inclusions, and alteration play a major role, often regardless of U content. Some extremely high-U zircons, which would be destroyed in no-time by CA, can provide concordant U-Pb data if they are just treated gently by air abrasion, demonstrating the relativity of these indicators. I would recommend that the authors reconsider and re-evaluate their discussion in merit.

Chemical abrasion by design leverages the fact that radiation-damaged zircon is more soluble than crystalline zircon. Establishing a relationship between radiation damage and zircon solubility is therefore fundamental to understanding how different chemical abrasion protocols affect zircon dissolution and U-Pb outcomes. Radiation-damaged zircon is also more susceptible to Pb loss and alteration than crystalline zircon. Consequently, understanding at what alpha dose zircon becomes susceptible to alteration and potential Pb loss is also a fundamental, outstanding question. The reviewer is correct that not all radiation-damaged zircon are affected by Pb-loss; the presence of fluids likely plays an important role. We will add this point in our revisions. The zircons analyzed in this study, however, are affected by Pb loss, so interrogating the threshold alpha dose at which Pb loss effects are apparent has merit. Future contributions by the U-Pb community will help determine whether the threshold alpha dose established for our samples is relevant to other zircon populations. We do not ignore the effects of textures, inclusions, and alteration on chemical abrasion or U-Pb data. In this paper and our GChron companion paper, "Chemical abrasion: The mechanics of zircon dissolution" we devote extensive discussion to the role of zonation, fractures, textures, and inclusions affect zircon dissolution on chemical abrasion and chemical and isotopic analyses – the alpha dose relationship is only one piece of chemical abrasion puzzle.

There seems to be some confusing concerning the parameters used in the various calculations for data from the literature, such as the 238/235 ratio. I suggest adding a table listing the original information, and the equivalent values calculated with the same constants as in the present paper. It would be useful for the reader, but also a reminder for the authors, avoiding comparisons of apples and oranges.

The $^{206}Pb/^{238}U$ and $^{207}Pb/^{206}Pb$ ages reported for AS3 by Schoene et al. (2006) were calculated assuming a U ratio of 137.88, whereas our ages assume a U ratio of 137.818. We will recalculate the literature data using a U ratio of 137.181 in revisions as the reviewer suggests for a more accurate comparison.

Line 30: We will make the suggested edit.

Line 32-34: We can rephrase this sentence, but we continue to conclude that this represents an important finding of this paper.

64: We will make the suggested edit.

129: We can rephrase these lines to improve clarity.

145-147: We can rephrase these lines to improve clarity.

177: We will make the suggested edit.

189-191: We will make the suggested edits.

224: Most of the Pbc in AS3 leachates is derived from inclusions and altered zones. The Pbc in the AS3 residues and 210 °C L2 & L3 leaching steps, however, is most likely derived from the blank.

264: The debate about the dome and keel structures of the Eastern Pilbara Craton is a debate over whether a stagnant lid or mobile lid tectonics regime was operating during the Archean.

274: The region likely remained at temperatures below ~460 °C based on the hornblende Ar-Ar, and apatite U-Pb (line 268).

Figure 6: WM stands for weighted mean. We will add this to the caption.

Figure 8: The error bars are smaller than the marker size. We will rephrase this.

343: The inclusions were not identified.

Figure 9: U ionization was very poor for these samples. We can add this to the text.

370: We will fix the text that the figure cut off.

373: 100 ppm U is fairly low for zircon.

378: We will fix the text that the figure cut off.

436: We agree with the interpretation of Takehara et al. (2018) that the altered zones reflect hydrothermal alteration. Low-temperature hydrothermal alteration is not uncommon. Fluids generally need to be present for alteration to occur.

465: Like the reviewer suggests, U concentration was estimated using the measured Th concentration and Th/U ratios. We can, however, include the estimated U concentration in the tables.

467: We will remove these lines as suggested.

522: We can rephrase this sentence to exclude the term frantzing.

598: We will remove the term discords.

649: We will replace "non-pattern" with "inconsistent behavior."

804: We will remove the duplicate reference.

---

## Author Response (AR1)

Our responses to the editor and reviewer suggestions are recorded below in blue.

**Editor Comments**

Thank you for your thorough responses to the reviewers' comments. Further stepwise dissolution experimental work on duration and temperature used in chemical abrasion and assessment of indicators (U, Pb*/Pbc, LREE concentrations, alpha-dosages of leachates and residues) for insights into zircon systematics is suited to this journal and will be of broad interest to the readership and of particular interest to the ID-TIMS community. Please submit a revised version considering each point the two reviewers have made and provide point-by-point responses to their comments.

Both reviewers had issues with the suggestion that knowledge of alpha dosage would be directly useful in 'tailoring' the CA procedure both from a practical point of view (Reviewer 1 notes that radiation dosage was calculated from the leachates after dissolution and that no information is given on how to best determine it on a population to help guide grain selection), and in light of the many factors that can influence discordance (Reviewer 2 notes that it is not realistic to take the degree of discordance to a direct function of alpha dosage when we know that texture/zonation, inclusions, alteration play a role). Please consider their comments when revising discussion of your results. The last sentence of the abstract "workers can better tailor the CA process to specific zircon…." seems to suggest that the paper will present a methodology and should perhaps be re-written.

We have removed the last sentence of the abstract, and we have revised the discussion regarding the use of alpha dose for "tailoring" chemical abrasion to specific zircon datasets. We have added points to the discussion highlighting the challenge of determining alpha dose prior to dissolution. We note that Raman and LA-ICPMS can be used to estimate alpha dose, however, both methods are resource- and time-intensive. No method presently exists for estimating radiation damage in 3D. We direct readers to Fig. 13 as a quick–and-dirty way of roughly estimating alpha dose given a range of typical U concentrations and an estimated damage accumulation interval. The final paragraph of the discussion highlights the role of crystal-specific factors and directs readers to our companion manuscript that discusses the effects of inclusions, alteration, and zonation on zircon dissolution at length.

Please also consider the numerous suggestions of Reviewer 2 pertaining to the data tables. In particular descriptions of the zircon grains in the tables would be useful given one of the aims of the study is to help understand systematics in light of grain characteristics.

We have updated all the data tables, and we have included CL and/or BSE images of all dated grains to the supplementary materials to provide readers additional information about grain characteristics.

When revising please remove repetitious text (for example conclusions lines 677-679) and as

noted by Reviewer 1, reorganization in the Results section could aid in comparing different zircon results while reducing repetition.

We have reorganized the Results section as suggested. We have moved three of the somewhat repetitious figures (the REE spider diagrams) to the supplementary materials.

**Reviewer #1**

The manuscript "Geochronological and Geochemical Effects of Zircon Chemical Abrasion: Insights from Single- Crystal Stepwise Dissolution Experiments" by McKanna and others is presenting step leaching U-Pb ID-TIMS data on three different zircon reference materials for two different temperatures and giving a recommendation for conditions to use by practitioners of chemical abrasion and an estimated minimum alpha dose at which Pb loss can occur. This study is a follow up from a previous study by the same authors on the same zircon samples that looked at structure of individual zircon grains of in response to chemical abrasion at different temperatures.

The conclusions from this study are largely confirming what has already been published about the chemical abrasion conditions and its consequences (Huyskens et al 2016, Widmann et al. 2019). All three studies concluded that leaching at 180C is not high enough for most samples to effectively remove all Pb loss from a sample and a higher temperature is needed in most cases. This study, however, is the only one that has detailed structural observations of zircons before and after the leaching procedure and it can tie them together with the U-Pb observations. This also leads to the possibility of estimated radiation damage of the leached zones, which could not be done in the two previous studies.

This is the main new angle of the study and these observations and discussions are unfortunately falling short in favour of documenting observations at length that have been made before. Thus, my recommendation is to focus a lot more on the new aspects of the study. To cut back on the repetitions reorganising of the results section would help. For example, for each sample early leachates are enriched in LREE. Thus keeping the standard format of first heading "geologic setting and sample description" with a sub heading of the different samples is advantageous. It also makes it easier to compare and contrast the different zircons analysed in this study.

We have reorganized the Results section to make it more concise and less repetitive.

The authors also looked for a tool to robustly identify zircons that have remaining Pb loss. Currently the identification of such analysis is very subjective and such a tool would make interpretations of a scattered dataset more robust. Unfortunately, neither of the indicators (U concentration, Pb*/Pbc, or LREE enrichment) are effective tools for this task.

Yes, the result is unfortunate, but important to demonstrate.

One of the recommendations of this paper is to look for the amount of radiation damage and tailor the leaching conditions this way. However, no information is given on how to determine the radiation damage prior to dissolution. In this study the radiation dose was calculated based on the analysis of U and Th in the leachate and residue of the zircons, which means after already performing the time-consuming analyses. In the prior study Raman was used to estimate the alpha dose for the zircons. Any method to determine the alpha dose would need to have a high spatial resolution for the entire volume of the zircon grain, since it was documented in the previous paper that these zones do exist also in the interior of some zircon grains.

As discussed above, we have revised the discussion regarding "tailoring" chemical abrasion. We have added points to the discussion highlighting the challenge of determining alpha dose prior to dissolution. We note that Raman and LA-ICPMS can be used to estimate alpha dose, however, both methods are resource- and time-intensive. No method presently exists for estimating radiation damage in 3D. We direct readers to Fig. 13 as a quick–and-dirty way of roughly estimating alpha dose given a range of typical U concentrations and an estimated damage accumulation interval.

Please provide all the calculated numbers that are used in the plots like alpha dose at Pb loss or LREE-I. – Values are included in Table S3.

A description is needed how the amount of dissolved material for the calculation of alpha dose was determined. The revised manuscript includes the alpha dose equation. The amount of dissolved material is not required for this calculation.

Fig 12: could you distinguish between the different temperatures maybe using open vs filled symbols? – We have made the suggested change (now Fig. 9).

Fig 15: It looks like the samples leached at 210C overall had lower radiation damage compared to the ones used in the 180C experiment in this figure and this would need an explanation. My guess is that it has something to do with the fraction of material in each of the dissolution steps, but this is confusing at first. For me, the figures about the alpha dose are the most important in this study and the groupings in Fig 14 and 15 are so broad that they could be masking interesting details. A plot including alpha dose vs discordance for example on an individual analysis basis could be interesting.

The reviewer is correct that the apparent differences in alpha dose between the samples leached at 180 °C and 210 °C reflect the fraction of material dissolved in each step. We have added this point to the revised discussion. In general, we find considering the data in aggregate to be meaningful for deriving overarching trends. Alpha does vs. discordance is not a useful metric for evaluating the AS3 and the KR18-04 datasets, since the data spread along the concordia line (Fig. 1 and Fig. 9).

Line 31:" However, Ultimately,…" – Corrected.

Line 49: "since the trajectory of Pb-loss follows Concordia" I think it should be the concordia. – Corrected.

Lines 50-51: "the precision of 207Pb/235U dates is also lower than corresponding 238U/206Pb dates due to the shorter radioactive half-life of 235U and lower isotopic abundance (Corfu, 2013; Schoene, 2014)." The precision of the 207Pb/238U dates is lower not because of the shorter half-life. It is only lower due to the lower abundance, which in turn is due to the shorter half-life. – Corrected.

Line 59:" annealing zircon samples prior to leaching helps to minimize the unwanted isotopic fractionation effects that plagued earlier leaching attempts" The main improvement is reducing elemental fractionation in the leaching steps. – We have added this to the revisions.

Line 191: "included and altered grains…" Grains with inclusions – Corrected.

Line 210: "which had been redone as part of (Schoene et al., 2006)."   Brackets are in the wrong place – Corrected.

Line 522: frantzing should not be a verb – Corrected.

**Reviewer #2 – Fernando Corfu**

The paper reports the results of experiments on the chemical abrasion of zircon, using 3 samples of different age, and applying two different experimental protocols. The reported results include U-Pb isotopic ratios and ages, and abundances of a number of chemical element typical in zircon. The experiments evaluate the efficiency of the two different approaches in removing discordant zircon domains and isolating grains with closed isotopic systems.

The data will be interest to the geochronologists that use the U-Pb dating methods, especially the ID-TIMS community. It is not the first paper to report such experiments, but it adds some new perspectives, which will certainly be useful for a further advancement of the technique.

The paper is reasonably well prepared. There are some technical glitches, with figures inserted in the text and locally covering up portions of the text. I have put some suggestions and a number of comments directly in the annotated file.

We have addressed the technical glitches. Responses to specific line edits are listed below.

The tables need some work to make them more useful and accessible for the readers. (1) It would be practical to assemble them all as separate sheets in just one file. (2) It would be practical to list U abundances, since they are mentioned repeatedly in the text. At present one has to use the Th/U ratios from one table and combine them with the Th abundances in another table to get an idea of the U contents. (3) There is no explanation of what (ppt) stands for (part per trillion, or per ton?), and ppt of what? Some solution? Because of this enigma the

listed numbers do not mean anything directly. Further back in the table there are then absolute abundances in ppm. Please, put those in the front, and explain all the terms used. (4) Please list the 206/204 ratios.

We have updated all the supplementary data tables accordingly. We have listed results for the different zircon samples in separate tabs in one U-Pb (Table S1) and one ICPMS file (Table S2). We added U concentrations to Table S2. 3) We moved the concentration of the element in zircon (ppm – parts per million) to the front Table S2. The concentration of the element in 1 mL solution (ppt – parts per trillion) is listed at the back of Table S2. We added the $^{206}Pb/^{204}Pb$ ratios to Table S1.

(5) The outcome of the experiments depends very strongly on the qualities and characteristics of the zircon grains used, but the tables provide no information in merit at all. One may perhaps try to link the individual data to the information in the previous associated paper by these authors. I highly recommend putting a characterization of each grain in the table. Geochronologists know that no two zircons are born alike, and they know that successful dating is best done by a strict discrimination of the good from the bad. A lack of information on the tested grains strongly weakens the interpretations and lessons learned from the study.

We have added CL and/or BSE images of dated grains to the Supplementary Materials (Fig. S1, S2, and S3). We feel this additional data will be more useful for evaluating the characteristics of dated grains than written descriptions in a table would. The general characteristics of the three zircon populations are discussed in detail in our GChron companion paper: "Chemical abrasion: The mechanics of zircon dissolution."

The discussion comprises are section linking a-dosage and degree of discordance, and its implications for the CA-application. It is certainly true that a-dosage and the relative radioactive damage are important factors affecting discordance. But it is also very simplistic, and not realistic, to reduce the degree of discordance to a straight function of a-dosage. Clearly, the textural factors, inclusions, and alteration play a major role, often regardless of U content. Some extremely high-U zircons, which would be destroyed in no-time by CA, can provide concordant U-Pb data if they are just treated gently by air abrasion, demonstrating the relativity of these indicators. I would recommend that the authors reconsider and re-evaluate their discussion in merit.

Chemical abrasion by design leverages the fact that radiation-damaged zircon is more soluble than crystalline zircon. Establishing a relationship between radiation damage and zircon solubility is therefore fundamental to understanding how different chemical abrasion protocols affect zircon dissolution and U-Pb outcomes. Radiation-damaged zircon is also more susceptible to Pb loss and alteration than crystalline zircon. Consequently, understanding at what alpha dose zircon *becomes susceptible* to alteration and potential Pb loss is also a fundamental, outstanding question. The reviewer is correct that not all radiation-damaged zircon are affected by Pb loss; the presence of fluids likely plays an important role. We have added this point in our revisions. The zircons analyzed in this study, however, are affected by Pb loss, so interrogating

the threshold alpha dose at which Pb loss effects are apparent has merit. Future contributions by the U-Pb community will help determine whether the threshold alpha dose established for our samples is relevant to other zircon populations. We highlight the role of crystal-specific factors such as inclusions, fractures and zonation in the revised manuscript and direct readers to our GChron companion paper, "Chemical abrasion: The mechanics of zircon dissolution" that is devoted to the extensive discussion of the role of zonation, fractures, textures, and inclusions on zircon dissolution.

There seems to be some confusing concerning the parameters used in the various calculations for data from the literature, such as the 238/235 ratio. I suggest adding a table listing the original information, and the equivalent values calculated with the same constants as in the present paper. It would be useful for the reader, but also a reminder for the authors, avoiding comparisons of apples and oranges.

The $^{206}Pb/^{238}U$ and $^{207}Pb/^{206}Pb$ ages reported for AS3 by Schoene et al. (2006) were calculated assuming a zircon U ratio of 137.88, whereas our ages assume a zircon U ratio of 137.818. We cannot easily correct legacy isotope ratios given the updated zircon U ratio without raw data. We have, however, calculated a new weighted mean $^{207}Pb/^{206}Pb$ date from the published isotope ratios assuming a zircon U ratio of 137.818 in the revised manuscript.

Line 30: Corrected.

Line 32-34: We have rephrased this sentence, but we continue to conclude that this represents an important finding of this paper.

64: Corrected.

129: We have drephrased these lines to improve clarity.

145-147: We have rephrased these lines to improve clarity.

177: Corrected..

189-191: Corrected..

224: Most of the Pbc in AS3 leachates is derived from inclusions and altered zones. The Pbc in the AS3 residues and 210 °C L2 & L3 leaching steps, however, is most likely derived from the blank.

264: The debate about the dome and keel structures of the Eastern Pilbara Craton is a debate over whether a stagnant lid or mobile lid tectonics regime was operating during the Archean. We have added this point.

274: The region likely remained at temperatures below ~460 °C based on the hornblende Ar-Ar, and apatite U-Pb (line 268).

Figure 6: WM stands for weighted mean. We added this to the figure caption.

Figure 8: The error bars are smaller than the marker size.

343: The inclusions were not identified.

Figure 9: U ionization was very poor for these samples. We add discussion to the text.

370: Corrected.

373: Corrected.

378: Corrected.

436: We agree with the interpretation of Takehara et al. (2018) that the altered zones reflect hydrothermal alteration. Low-temperature hydrothermal alteration is not uncommon. Fluids generally need to be present for alteration to occur.

465: We have included U concentration in Table S2.

467: We removed these lines as suggested.

522: Corrected.

598: Corrected.

649: Corrected.

804: Corrected.